



# 1  A reexamination of the dry gets drier and wet gets wetter paradigm
# 2  over global land: insight from terrestrial water storage changes

Jinghua Xiong[1]; Shenglian Guo[1*]; Jie Chen[1]; Jiabo Yin[1]
[1] State Key Laboratory of Water Resources and Hydropower Engineering Science, Wuhan University, Wuhan, 430072,
China
*Correspondence to*: S. Guo (slguo@whu.edu.cn)
**Abstract.** The "dry gets drier and wet gets wetter" (DDWW) paradigm has been widely used to summarize the expected
trends of the global hydrologic cycle under climate change. However, the paradigm is challenged over land due to different
measures and datasets, and is still unexplored from the perspective of terrestrial water storage anomaly (TWSA).
Considering the essential role of TWSA in wetting and drying of the land surface, here we built upon a large ensemble of
TWSA datasets including satellite-based products, global hydrological models, land surface models, and global climate
models to evaluate the DDWW hypothesis during the historical (1985-2014) and future (2071-2100) periods under various
scenarios. We find that 27.1% of global land confirms the DDWW paradigm, while 22.4% of the area shows the opposite
pattern during the historical period. In the future, the DDWW paradigm is still challenged with the percentage supporting the
pattern lower than 20%, and both the DDWW-validated and DDWW-opposed proportion increase along with the
intensification of emission scenarios. Our findings will provide insights and implications for global wetting and drying
trends from the perspective of TWSA under climate change.

## 18  1 Introduction

The hydrological conditions of the land surface have experienced considerable changes due to climate change and
anthropogenic interventions, exerting a tremendous impact on regional agriculture, ecological environment, and fresh
availability (Shugar et al., 2020; Gampe et al., 2021). Assessing the variations of terrestrial wetness and dryness is crucial in
understanding the hydrological response and dealing with water-related issues in the context of global change (Moreno-
Jimenez et al., 2019; Zhao et al., 2021). Under these circumstances, the 'dry gets drier and wet gets wetter' (DDWW)
paradigm was developed based on the deficit between precipitation and evapotranspiration (P − E). This deficit is expected
to enhance due to increase in atmospheric water vapor in humid regions under a warming climate, and vice versa (Durack et
al., 2012). The DDWW paradigm has been frequently used to represent the historical and future trends in hydrologic cycle
under climate change in both regional and global scales (Held and Soden, 2006; Chou et al., 2009; Allan et al., 2010; Donat
et al., 2016; Hu et al., 2019; Zeng et al., 2019). However, the rational of DDWW mechanism is recently questioned at
different levels through the growing accessibility of datasets, models, and indicators (Greve et al., 2014; Byrne and



O'Gorman, 2015; Greve and Seneviratne, 2015; Kumar et al., 2015; Polson and Hegerl, 2017; Yang et al., 2019; Y. Li et al.,
2021). Byrne and Gorman (2015) used simulations from 10 climate models to reveal an ocean-land contrast pattern in the
response of $P - E$ to global warming in historical (1976-2005) and future (2071-2099) periods, highlighting the DDWW
mechanism more suitable over ocean than over land. Given the fact that historical evaluation of the DDWW paradigm is
mainly based on oceanic records, Greve et al. (2014) adopted 2142 possible combinations of $P - E$ to assess the trends in
wetting and drying over global land and discovered merely 10.8% of area following the DDWW pattern during the period
1948-2005. Alternatively, Yang et al. (2019) integrated an ensemble of six hydro-climatic indicators for the global
assessment of the DDWW paradigm between 1982 and 2012, suggesting the catchphrase only occurred over 20% of the
global land. In short, there are great uncertainties still remaining in the assessments of global trends in dryness and wetness
under climate change (Dai, 2011; Trenberth et al., 2014).
The uncertainties within previous studies are mainly sourced from different choices of measures and datasets for
changes in dryness and wetness (Vicente-Serrano et al., 2010; Feng and Zhang, 2015; Huang et al., 2016). Specifically, the
widely used metric $P - E$ over ocean has been proven overwhelmingly positive over land based on both observations and
simulations (Greve et al., 2014; Byrne and O'Gorman, 2015; Greve and Seneviratne, 2015). Meanwhile, some indices
derived from precipitation and evapotranspiration like the standardized precipitation evapotranspiration index (SPEI), aridity
index (AI) and standardized precipitation/evapotranspiration index (SPI/SETI) neglect the hydrological process on the land
surface (Huntington, 2006; Dai, 2011). In addition, a few indexes like the standardized soil moisture index (SSI),
standardized groundwater index (SGI), and standardized runoff index (SRI) merely highlight differently single aspect of the
water cycle, lacking the complete representation of the terrestrial water storage (TWS) (AghaKouchak, 2014; Wu et al., 2018;
Guo et al., 2021). It is largely worth noting that TWS consisting of water storage in surface water, soil moisture,
groundwater, snow and ice, and canopies can physically provide integrated information about the overall status of land,
whose changes are closely linked to the terrestrial wetting and drying tendency (Tapley et al., 2019; Pokhrel et al., 2021).
Therefore, there are several TWS-based indicators showing effective ability for monitoring and assessment of large-scale
droughts like the total storage deficit index (TDSI) and floods such as the flood potential index (FPI), indicating an urge
necessary to examine the DDWW paradigm from the perspective of terrestrial water storage changes (Long et al., 2014;
Xiong et al., 2021a). Alternatively, divergent data sets produce different trends due to distinctive internal variability and
external forcing (from satellites and meteorological stations), especially for precipitation and evapotranspiration (Chen et al.,
2020). Moreover, Scanlon et al. (2018) conducted comprehensive comparisons between decadal trends in TWS from seven
global models and three Gravity Recovery and Climate Experiment (GRACE) satellite solutions over major basins globally,
a great underestimation of the increasing and decreasing trends of models was discovered due to human water use and
climate variations. Generally, a systematic reexamination of the DDWW paradigm from the perspective of terrestrial water
storage anomalies (TWSA) is necessary yet still lacking at the global scale.
Therefore, we use an ensemble of 9 different TWS datasets from the GRACE reconstructions, global hydrological
models, and land surface models to examine the DDWW paradigm over the global land between 1985 and 2014.



Subsequently, an alternative ensemble of 8 global climate models (GCMs) from the Coupled Model Intercomparison Project
6 (CMIP6) is used to further test the paradigm under various scenarios during the 2071-2100 period.

## 2 Data and Methods

### 2.1 Data

We used an ensemble of 9 data sets (hereinafter "DATASET") to evaluate the DDWW paradigm during the historical
period 1985-2014, which includes 3 GRACE reconstructions, 3 global hydrological models, and 3 global land surface
models (see Table S1). The ensemble mean was estimated using a simple average method to eliminate the uncertainty among
different datasets. All of the members and their mean has been resampled to 1° × 1° grid cell for comparing with the average
value of three GRACE mass concentration blocks (mascon) solutions between June 2002 and December 2014, during which
the missing months of GRACE have been filled using a linear interpolation method. Alternatively, an ensemble of 8
simulations from CMIP6 was used to examine the DDWW paradigm in the future period (2071-2100). All the ensemble
members have been resampled to 1° × 1° scale using a bilinear interpolation approach for better comparison in spatial
domain. Similarly, the ensemble mean of CMIP6 models has been estimated using simple averaging. It is worth noting that
all the DATASET and CMIP6 members and their ensemble means have been removed from the base line between 1985 and
2014 to be consistent with GRACE data.

### 2.1.1 GRACE and GRACE Reconstructions

A total of 3 GRACE reconstructions provided by Humphrey and Gudmundsson (2019) and Li et al. (2021) were
selected to join the DATASET for evaluation of the DDWW paradigm. The ensemble of GRACE reconstructions is
generated based on the stat-of-the-art machine learning models using historical and near-real-time meteorological forcing. It
is informative to note that the GRACE reconstructions from the Humphrey and Gudmundsson (2019) were calibrated with
GRACE mascon solutions from NASA JPL and NASA Goddard Space Flight Center (GSFC), respectively, and that
supplied by Li et al. (2021) was trained by the GRACE mascon product from Center for Space Research (CSR). The
accuracy and applicability of three GRACE reconstructions has been fully evaluated over global land in several previous
studies (Xu et al., 2021; Yi et al., 2021). Correspondingly, three latest GRACE mascon solutions (RL06-v02) from JPL,
GSFC, and CSR were prepared for comparison purpose. The mascon solutions are believed to be more reliable than
traditional spherical harmonic products due to lower signal leakage and less post-processing dynamics (Xiong et al., 2021b).

### 2.1.2 Global Hydrological Models

We used three global hydrological models including the Variable Infiltration Capacity macroscale model (VIC-v4.1.2),
the WaterGAP hydrological model (WGHM-v2.2d), and PCRaster GLOBal Water Balance model (PCR-GLOBWB-v2.0) to



estimate TWS for evaluation of the DDWW paradigm. The offline physically based, semi-distributed, grid-based VIC model is managed by the NASA Global Land Data Assimilation System Version 2.1 (GLDAS-v2.1) (Liang et al., 1994; Syed et al., 2008). Forced by the Global Data Assimilation System atmospheric analysis fields (Derber et al., 1991) and the Air Force Weather Agency's AGRicultural METeorological modeling system radiation fields, the VIC model can effectively capture the terrestrial water cycle by simulating the water stored in the canopies, snow, and soil moisture within the depth of 3 soil layers (200 cm). The VIC model has been widely used to analyze the terrestrial water storage changes at both regional and global scales (Hao and Singh, 2015; Hao et al., 2018). The WGHM is a grid-based global hydrological model quantifying the human water use and continental water fluxes for all land areas excluding the Antarctica (Müller Schmied et al., 2021). Unlike most global hydrological models, the WGHM forced by the ERA40 and ERA-Interim reanalysis is able to simulate the groundwater storage by coupling with global water use models like the Groundwater-Surface Water Use, suggesting a comparably better representation of TWS (Döll et al., 2014). Several frequently-used model output such as TWS, stream, and water use have been evaluated against global observations (Wan et al., 2021). The PCR-GLOBWB model is a grid-based global-scale hydrology and water resources model that fully integrates water use such as water consumption, water withdrawal, and return flows (Sutanudjaja et al., 2018). Forced with the EC-Earth data including atmospheric, oceanic, and, land surface variables, the PCR-GLOBWB model can simulate the entire terrestrial surface over global land at a daily time scale. The model performance has been fully evaluated using global discharge measurements and supported many TWS studies globally (Scanlon et al., 2018; van der Wiel et al., 2019).

### 2.1.3 Land Surface Models

We used 3 land surface models consisting of the Noah (-v3.6), Catchment (CLSM-vF2.5), and CPC (-v2) models to calculate TWS globally for assessment of the DDWW paradigm. The Noah and CLSM models are managed by GLDAS (v-2.1) from the NASA GSFC institute. GLDAS is a suit of global hydrological models and land surface models that modeling optimal fields of land surface by integrating multi-source observations such as in situ stations and satellites based on start-of-the-art data assimilation and land surface simulation techniques (Rodell et al., 2004). GLDAS has been widely used to compare with GRACE TWSA in data-sparse regions such as Africa and Qinghai-Tibetan Plateau (Ogou et al., 2021; Xing et al., 2021). The Noah-modeled TWS is considered as the sum of canopy water storage, snow water equivalent, and soil moisture of four layers with a depth of 200 cm. Different from that, the CLSM simulates the shallow groundwater and the vertical levels of soil moisture is not explicitly divided within the depth of 100 cm. Developed by the U.S. National Oceanic and Atmospheric Administration (NOAA), the CPC model provides global soil moisture conditions in 160 mm column soil forced with observations of different metrological and hydrological fluxes (e.g., precipitation, temperature, and humidity) from CPC (Fan, 2004). The reasonably good ability of CPC simulations to capture the TWS dynamics has been examined over many areas of the globe in spite of its simplicity in the calculation of TWS (Jin et al., 2012; Agutu et al., 2020).



### 2.1.4 Global Climate Models

We used a suit of 8 global climate models belonging to the ensemble "r1i1p1f1" of CMIP6 to evaluate the DDWW paradigm under climate change. The CMIP6 serves as a category of experiments of global climate models coupled to a dynamic ocean, a simple land surface and thermodynamic sea ice (Eyring et al., 2016). The CMIP6 comparisons have become a diagnosis tool to better understand climate change in past, present, and future period (Krishnan and Bhaskaran, 2020). CMIP6 includes a total of five Shared Socio-economic Pathway (SSPs) representing global economic and demographic changes under different greenhouse gas emissions. We selected 3 out of 5 SSP scenarios including SSP126, SSP245, and SSP585, representing taking the green roads, middle of the road, and the highway road, respectively (Iqbal et al., 2021). In specific, the monthly average TWS from CMIP6 is estimated as the sum of total soil moisture and snow water, which has been proven reliable to assess the TWS changes (Wu et al., 2021). To avoid the considerable uncertainties in TWS of different CMIP6 models, a trend-preserving method was employed to perform bias correction combined with historical GRACE data. The trend-preserving method initially developed by Hempel et al. (2013) modifies the monthly means of the simulated data to match the observed results using a constant offset between the simulations and observations, and has been widely used in the Intersectoral Model Intercomparison Project (ISIMIP2b).

### 2.2 Detection of Wetting and Drying

The non-dimensional TWS drought severity index (TWS-DSI) was adopted to reflect the long-term trends in terrestrial dryness and wetness at both $1° \times 1°$ grid cell and regional scales over global land (see Figure S1 and Table S2). TWS-DSI has been widely used in hydrology and climate field due to its simple structure and effective ability in capturing drying and wetting condition (Pokhrel et al., 2021). Monthly TWS-DSI was calculated for all ensemble members and their mean from DATASET and CMIP6 as follows (Zhao et al., 2017):

$$TWS - DSI_{i,j} = \frac{TWS_{i,j} - \mu_j}{\sigma_j} \qquad (1)$$

where $TWS_{i,j}$ is the TWS value in year $i$ and month $j$; $\mu_j$ and $\sigma_j$ denotes the mean and standard deviation of the annual TWS in month $j$, respectively. Long-term trends in TDS-DSI was estimated using the simple linear regression method and the significance of trend values is evaluated using the t-test at a 5% significance level. The area having a significant trend of increasing/decreasing TWS-DSI is considered undergoing wetting/drying, otherwise is defined as an uncertain region.

### 2.3 Selected Regions

Our study performed the assessment of the DDWW paradigm over global land excluding the Greenland at both gridded $1° \times 1°$ cell and regional scales. A total of 43 regions are selected based on the Special Report on Extremes (SREX) regions from Intergovernmental Panel on Climate Change (IPCC), which covers all the land area except for the Greenland (see Figure S1). Their basic information is summarized in Table S2.



## 3 Results and Discussion

### 3.1 Global Trends of Dryness and Wetness

Prior to the detection of the DDWW paradigm, we performed the evaluation of TWSA derived from the ensemble mean of the DATASET and CMIP6 achieve. Figure S2 presents the global distribution of the normalized root mean square error (NRMSE) between GRACE TWSA and that from DATASET as well as CMIP6 data after bias correction during the period April 2002-December 2014, which is calculated as the ratio of RMSE to the change range of TWSA (Xiong et al., 2020). The NRMSE between GRACE and DATASET is generally lower than 0.3 (95.7%), of which 12.9% of grid cells showing NRMSE below 0.1 and the percentage is 77.0% for the NRMSE lower than 0.2. Relatively larger NRMSE ranging from 0.3 to 0.4 occurs in the east and central Asia, south Australia, north Africa, and northeastern America, indicating relatively poorer performance of global hydrological and land surface models. Results over the central America, east Europe and west Asia is mostly below 0.1. However, the NRMSE between GRACE and CMIP6 is generally higher than that of DATASET, most (98.2%) of which changes from 0.1 to 0.5. Approximate 78.2% of the grid cells show NRMSE<0.3 and 34.2% has the NRMSE<0.2. Spatially, there are some extreme values as high as 0.8 located in the central and east Asia, south Africa, west and east of America, and east Europe, probably arising from the uncertainties in the simulations of CMIP achieve. Figure S3 further presents the regional results over the 43 selected SREX regions. Most mid-latitude regions like the WCE, EEU, WSB, ESB, and RFE present relatively lower NRMSE (0-0.1) between GRACE and DATASET, suggesting poorer performance than that in the NZ, ECA, NEU, and NEN. For CMIP6 data, a similar pattern could be discovered that most mid-latitudes regions have relatively lower NRMSE values from 0.1 and 0.2, while higher values (0.5-0.6) are located in NZ, NES, WCE, ESAF, and MDC. Although the bias correction has been performed to the CMIP6 TWSA, comparatively large bias still exists owing to the uncertainty in parameter, meteo-hydrological forcing, and internal variability of GCMs. Such biases can potentially influence the assessment of the DDWW paradigm under in the future period (2071-2100) climate change.

Temporal comparison of global average TWSA derived from DATASET/CMIP6 and GRACE during the period 2002-2014 is shown in Figure S4. The GRACE TWSA changing from −20 to 20 mm shows obvious seasonal characteristics with relatively higher uncertainty in dry season than that in wet season. A similar change pattern is captured by the DATASET, with the change range covering the variations of GRACE data. The NRMSE between the ensemble mean of DATASET and GRACE data is 0.11, lower than that (0.28) between the ensemble mean of CMIP6 and GRACE results. Moreover, the fluctuation range of CMIP6 data is generally slighter than the DATASET, highlighting the effective bias correction performance.

To provide insights in the aspect of terrestrial water storage changes for the evaluation the DDWW paradigm, TWS-DSI is estimated to determine the terrestrial wetness and dryness. Figure 1 shows the global distribution of long-term trends in TWS-DSI over the historical period 1985-2014 and future period 2071-2100 under SPSP126, SSP245, and SSP585 scenarios. During the historical period, a clear spatial divergence is observed globally and the average TWS-DSI has a significant decreasing slope of −0.07/a (*p*<0.05), similar to the results from SPI, SPEI, and AI (Wang et al., 2018; Yang et al.,





2019). Spatially, severe drying exists in the Gulf of Alaska coast and the Canadian archipelago with significant slopes of
TWS-DSI ranging from −0.08/a to −0.12/a, which is caused by rapid ice-sheet and glacier ablation under a warming climate
(Luthcke et al., 2013; Velicogna et al., 2014). Trigged by historical severe droughts over decades, the drying trends in central
Canada, southern California and Texas can be clearly discovered, among which the decreasing speed of TWS-DSI ranges
from −0.04/a to −0.12/a ($p<0.05$) (Bouchard et al., 2013; Haacker et al., 2016), so as the eastern Brazil (Getirana, 2016).
Moreover, overwhelming groundwater depletion due to unsustainable human water use such as irrigation is responsible to
the declining dryness at significant slopes ranging from −0.12/a to −0.16/a in southeast and north Africa, south Europe,
North China Plain, and northern India (Rodell et al., 2009; Feng et al., 2013; Ramillien et al., 2014). Naturally, moderate
drying trend in southwestern Africa caused by precipitation decrease is detected by the reduction of TWS-DSI. On the
contrary, increasing precipitation dominates the wetting trend in mid-latitude regions including southern Russia and
Canadian, west Africa, southeast Asia and Qinghai-Tibetan Plateau with significant slopes ranging from 0.04/a to 0.12/a
(Siebert et al., 2010; Ndehedehe et al., 2017). Alternatively, some regions such as Amazon River basin, south Africa and
eastern Australia presenting wetting trends are considered experiencing progression from wet to dry period (Chen et al.,
2010; Gaughan and Waylen, 2012).
In the future, most of the mid-latitude regions such as north China, south Mongolia, and central Europe are projected to
be wetting because of the growth in precipitation under SSP126 scenario (Milly et al., 2005; Seneviratne et al., 2006).
Similar trends can be found in North China Plain and Caspian regions that undergone drying during historical period due
mainly to groundwater abstraction and sporadic droughts. While some areas including the northern India and southwestern
America are expected to continue drying under SSP126 scenario in the future owing to the increasing evapotranspiration in a
warming climate (Allen et al., 2010; Vicente-Serrano et al., 2010). Alternatively, the obvious drying trend around the
Canada's subarctic lakes are attributed to the high vulnerability to droughts when snow cover declines under increasing
temperature (Bouchard et al., 2013). Many regions around the Aral Sea and north Russia are prone to experience wet-to-dry
progression under climate change. It is worthy noting that a higher emission scenario can be translated to a more intensive
trend of either drying or wetting, the pattern is also revealed by Pokhrel et al. (2021).
We further demonstrate the global distribution of the long-term trends in TWS-DSI over 43 selected SREX regions in
Figure 2. During the historical period, 54.1% and 45.9% of grid cells present drying and wetting trend, respectively, of
which 57.4% and 48.4% is significant ($p<0.05$). NAU has the highest percentage (73.9%) of grid cells with significant
increasing trend of TWS-DSI, which is mainly caused by precipitation increase (Rajah et al., 2014). While the ARP has the
greatest percentage of 82.0% of pixels showing significant drying trend jointly affected by the groundwater depletion and
droughts over the Arabian Peninsula (Lelieveld et al., 2012). During the future period under climate change, the proportion
of drying area with a significant slope increases from SSP126 (23.3%) to SSP585 (32.3%) scenario. A similar growth is
detected in the percentage with significant wetting trends, which reaches 15.6%, 17.8%, and 23.3% under SSP126, SSP245,
and SSP585 scenario, respectively. Some mid-latitude regions including WCE, EEU, WSB, WNA, and ECA present wetting
trend benefit from precipitation increase, and become wetter when a high emission scenario is expected to occur. SAS also



illustrates a dry-to-wet transformation along with a higher radiative forcing from SSP126 to SSP585. Alternatively, all the
regions in the north America and Russia except for WNA are expected to become drying, and so do some regions over the
southeastern Africa, central Asia and south of Australia. Generally, the percentage of grid cells showing significant trends of
both wetting and drying stably increase from the SSP126 to SSP585 scenarios, and the drying is always 10% higher than the
wetting.

**Figure 1: Global distribution of the long-term trends (left panel) and classification (right panel) in TWS-DSI during (a, b) the historical (1985-2014) and future (2071-2100) period under (c, d) SSP126, (e, f) SSP245, and (g, h) SSP585 scenarios. Note: The stippling marks regions with a significant trend. "D" and "W" indicates regions with drying and wetting trends, respectively.**

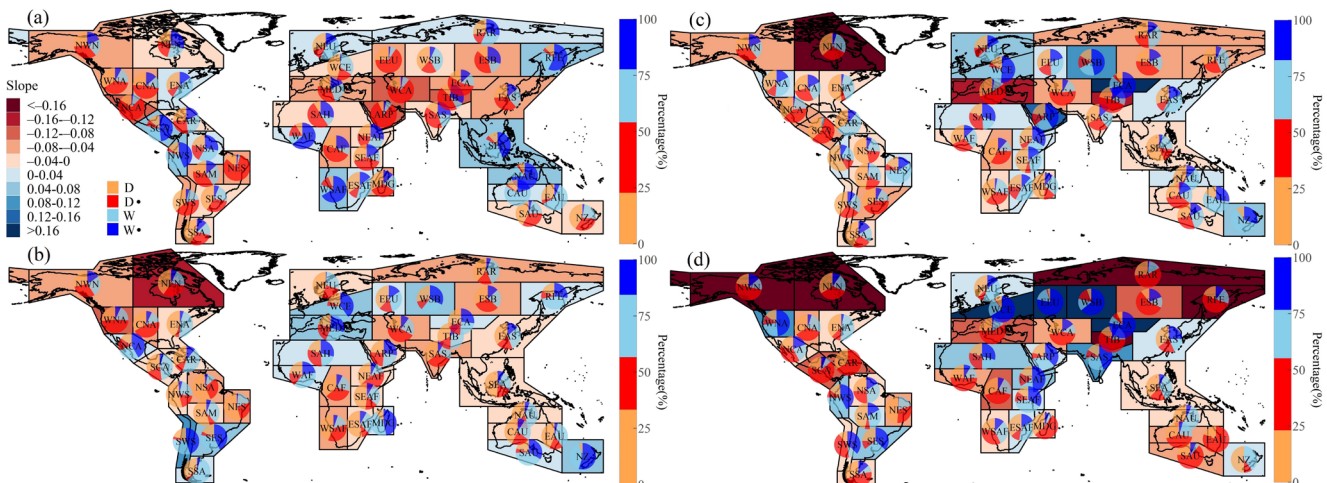

**Figure 2: Global distribution of the long-term trends in TWS-DSI and regional percentages in selected IPCC SREX regions during the (a) historical (1985-2014) and future (2071-2100) period under (b) SSP126, (c) SSP245, and (d) SSP585 scenarios. Note: The stippling marks regions with a significant trend. "D" and "W" indicates regions with drying and wetting trends, respectively.**

**3.2 Global Trends of Dryness and Wetness**

To evaluate the DDWW paradigm over global land, the effective Aridity index (AI) is used to classify a grid cell as arid, humid, and transitional region following Yang et al. (2019). The AI is calculated as the ratio of annual precipitation to potential evapotranspiration provided by the Climatic Research Unit Time series (CRU TS-v4.05) during the same period as TWS-DSI (i.e., 1985-2014). The global distribution of multi-year average AI as well as the classifications during the period 1985-2014 is presented in Figure S5. It can be seen that most arid regions are located in the southwestern America, north and south Africa, central Asia, Arabian regions, and Australia, accounting for 33.6% of the land. The percentage of humid area that mainly located in the east America, Amazon region, central Africa, south China, west Europe, and Russia reaches 58.1% of the land. Approximate 8.3% of land area is defined as the transitional region. We perform comparison between AI and TWSA derived from DATASET and CMIP6 between 1985 and 2014 in Figure S6, with the latter presenting similar spatial distribution to AI. Moreover, the CMIP6 data has a relatively higher amplitude than that of the DATASET, in line with the temporal results (see Figure S4).

Figure 3 illustrate the test results of DDWW paradigm at a 5% significance level during the historical and future period under climate change. A large quantity of grid cells over the north Africa, Arabian regions, east Asia, and southwest America show the DD phenomenon. In contrast to that, a substantial portion of area over the arid regions of the north and south of Africa, Australia, and central Asia shows the "dry get wetter" (DW) principle. Moreover, the WW paradigm is confirmed mainly in the east Russia and north Amazon, with the "wet get drier" (WD) pattern happen in the central Africa, northeastern Amazon, and north Asia. Under climate change, a similar pattern under SSP126 scenario is revealed to the historical results. Nevertheless, SSP245 scenario presents a slightly different distribution from historical results that many regions in the north Asia and central Europe show DW and WW situation instead of DD and WD. In addition to that, south



and northeast of China together with the majority of Russia show WD situation, and the DD paradigm is gradually dominating the Australia. This difference is further confirmed based on the results under SSP585 scenario. We further conducted the regional analysis as shown in Figure 4. Among 29.9% of land where showing significant trends in drying and wetting, 50.1% confirms the DDWW paradigm, of which 27.4% and 22.7% is drying and wetting, respectively during the historical period. ARP has the highest percentage of 96.4% of area with significant trends showing the DD hypothesis, while RFE achieves the highest proportion of 93.0% presenting the WW theory. During the future period under climate change, multiple SSP scenarios highlight a generally consistent distribution with some difference. Some regions such as WCA, WNA, and MED show a DD paradigm. In addition, WCE, SES, MDC, and NC has a WW paradigm. Apart from that, a lot of regions located in Russian, Mongolia, and Canada present WD condition, with few areas like NEAF and ARP showing DW situation. Generally, it can be clearly observed that the proportion of regions showing WD pattern increase from SSP126 to SSP585 scenario, indicating the comprehensive trend of drying at region scale.

Global statistics of the regions with various patterns during the historical and future periods are shown in Figure 5. During the period 1985-2014, a percentage as high as 54.2% of area shows significant trends in wetting or drying ($p<0.05$). Further, 27.1% of the area shows the DDWW paradigm, in which 14.9% and 12.2% of area is drying and wetting, respectively. 22.4% of the area, however, show the opposite pattern of DW (7%) and WD 15.4%, respectively. The confirmed percentage for the DDWW paradigm of this study (27.1%) is more than twice as high as a previous study (10.8%) due to divergent measures, datasets, and study period (Greve et al., 2014). Feng and Zhang (2015) used soil moisture to conclude a proportion of 15.12% following the DDWW pattern while a percentage of 7.7% of the land showing an opposite pattern between 1979 and 2013, relatively lower than our study. Yang et al. (2019) applied a combined measure to evaluate the DDWW paradigm and discovered the percentage following and opposing the DDWW paradigm is respective 29% and 20% during the period 1982-2012, highly consistent with this study. Under climate change, the proportion of area supporting the DDWW paradigm is 17.7%, 17.0%, and 20.0% under SSP126, SSP245, and SSP585 scenarios, respectively. Alternatively, the area having the opposite DDWW pattern achieves 17.9%, 22.7%, and 30.3%, respectively. The percentage of area with significant wetting and drying trends slightly increases over the enhancement of emission intensity, which is possibly related to the increase of DDWW-validated areas from SSP126 to SSP585 scenarios. It is worthy noting that internal variability of climate models might affect the potential agreement with the DDWW pattern (Kumar et al., 2015). Greve and Senevirtne (2015) used climate projections from CMIP5 to establish the measure P − E for assessment of the DDWW paradigm and discovered the hypothesis was validated over 19.5% of land area between 2080 and 2100 under the RCP8.5 scenario, which is close to our result (20.0%). Moreover, Li et al. (2021) further tested the DDWW theory with the third phase Paleoclimate Modeling Intercom-parison Project (PMIP3) simulations, concluding a similar proportion of 22.81% of the global land to our study that held the DDWW paradigm.

Despite the evaluation of the DDWW at a 0.05 significance level, we further tested the sensitivity of this theory to the different choices of significance level (see Figure S7). At a significance level of 0.01, 21.75% of land area agrees well with the DDWW theory, while the 17.0% of area illustrates the opposite pattern during the period 1985-2014. As for the 0.1





significance level, the DDWW-validated regions account for the 30.1% of total area, with 24.4% of land agreeing with the
opposite hypothesis. In the future period, a similar pattern is discovered that the both DDWW-confirmed and DDWW-
opposed regions are increasing along with the enhancement of projected strength of radiative forcing, with the reduction of
area showing insignificant trends in wetting and drying. However, the magnitudes of results at 0.01 significance level are
generally lower than that at 0.1 significance level due to the different thresholds of detected trends in drying and wetting.
Alternatively, considering the substantial uncertainties sourced from different models and scenarios, another sensitivity
analysis is conducted at individual member level (see Figure S8). For the historical period, a clear overestimation of the CSR
reconstructions is detected with 40% of area agreeing with the DDWW pattern, and 38.4% showing the opposite situation.
Moreover, the modeled results from VIC and WGHM illustrate the underestimation of the area validating the DDWW
paradigm, reaching 15.7% (WGHM) and 11.8% (VIC), respectively. Their proportion with the opposite DDWW paradigm is
9.8% (WGHM) and 17.8% (VIC), respectively. Therefore, it can be concluded that the differences among different members
of DATASET limitedly affect the evaluation of the DDWW during the historical period. In the future, three climate change
scenarios share a similar pattern that the GFDL-ESM4 model presents overestimation but the IPSL-CM6A and CanESM5
models have underestimation for different percentages. For specific, area dominated by the DDWW paradigm changes from
8.8% (CanESM5) to 21.4% (GFDL-ESM4), while that showing the opposite pattern ranges from 8.6% (CanESM5) to 15.4%
(GFDL-ESM4) under the SSP126 scenario. For the SSP245 scenario, the DDWW-validated regions account from 7.5%
(CanESM5) to 20.8% (GFDL-ESM4), the opposite pattern occurs over a range from 9% (CanESM5) to 16.3% (GFDL-
ESM4) of land. The proportion supporting the DDWW paradigm fluctuates from 9.9% (CanESM5) to 23.7% (GFDL-ESM4),
while that presenting the opposite pattern ranges from 8.2% (CanESM5) to 13% (GFDL-ESM4) under the SSP585 scenario.
Overall, the comparatively large difference among various models might source from unforced internal climate variability of
distinctive CMIP6 members. Uncertainties sourced from different emission scenarios might be another issue (Kumar et al.,
310  2015).

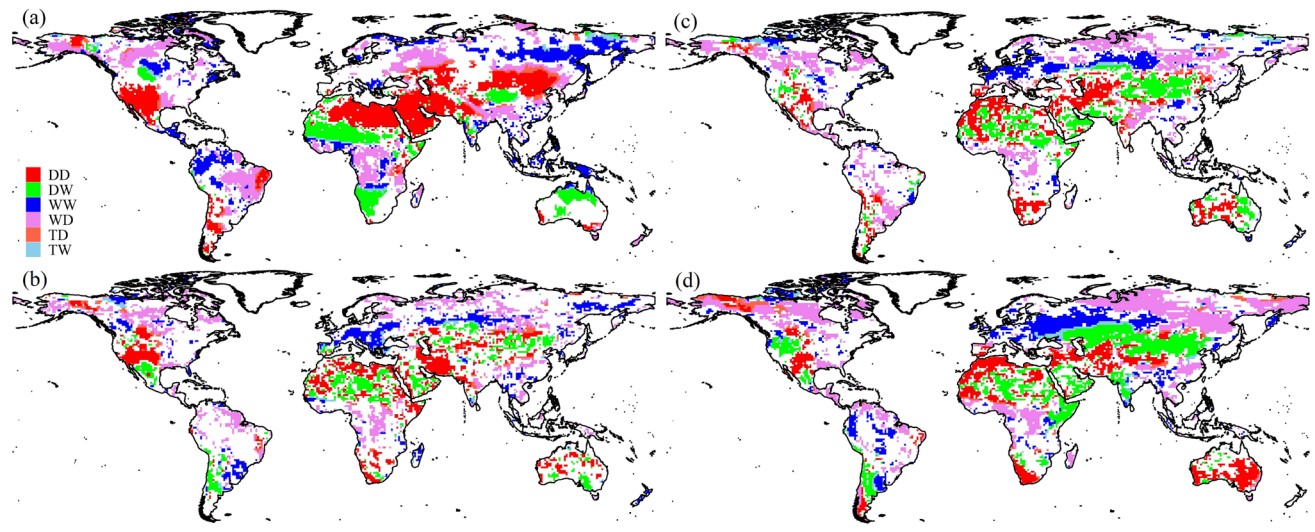





**Figure 3: Global assessment of the DDWW paradigm during the (a) historical (1985-2014) and future (2071-2100) period under (b) SSP126, (c) SSP245, and (d) SSP585 scenarios. Note: The DD indicates the dry gets drier; the DW indicates the dry gets wetter; the WW indicates the wet gets wetter; the WD indicates the wet gets drier; the TD indicates the transition gets drier; the TW indicates the transition gets wetter.**

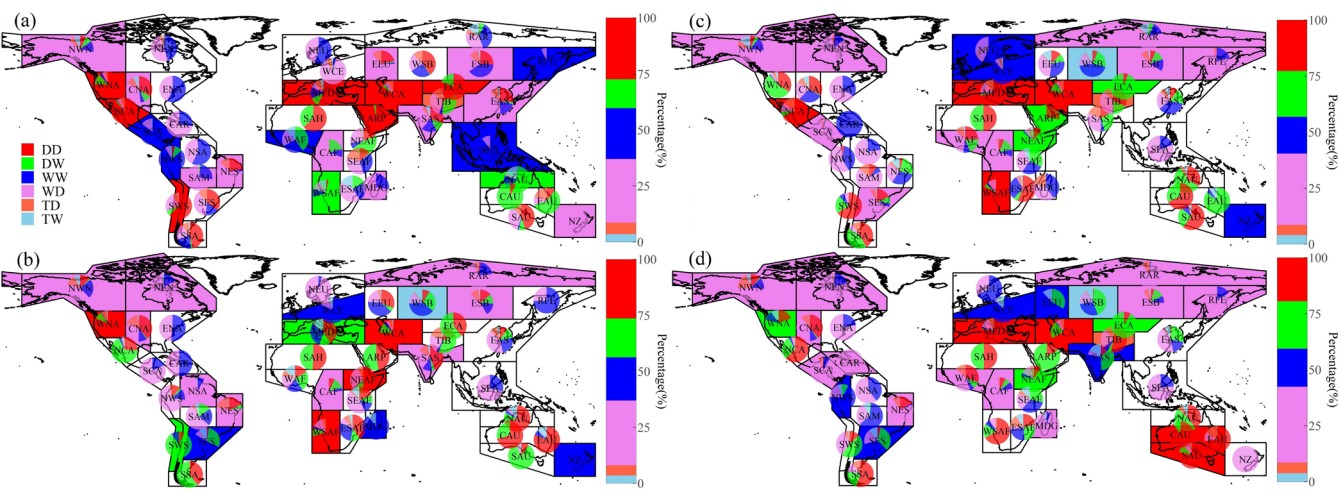

**Figure 4: Global assessment of the DDWW paradigm and the regional percentages of area with significant trends in selected IPCC SREX regions during the (a) historical (1985-2014) and future (2071-2100) period under (b) SSP126, (c) SSP245, and (d) SSP585 scenarios. Note: The DD indicates the dry gets drier; the DW indicates the dry gets wetter; the WW indicates the wet gets wetter; the WD indicates the wet gets drier; the TD indicates the transition gets drier; the TW indicates the transition gets wetter.**

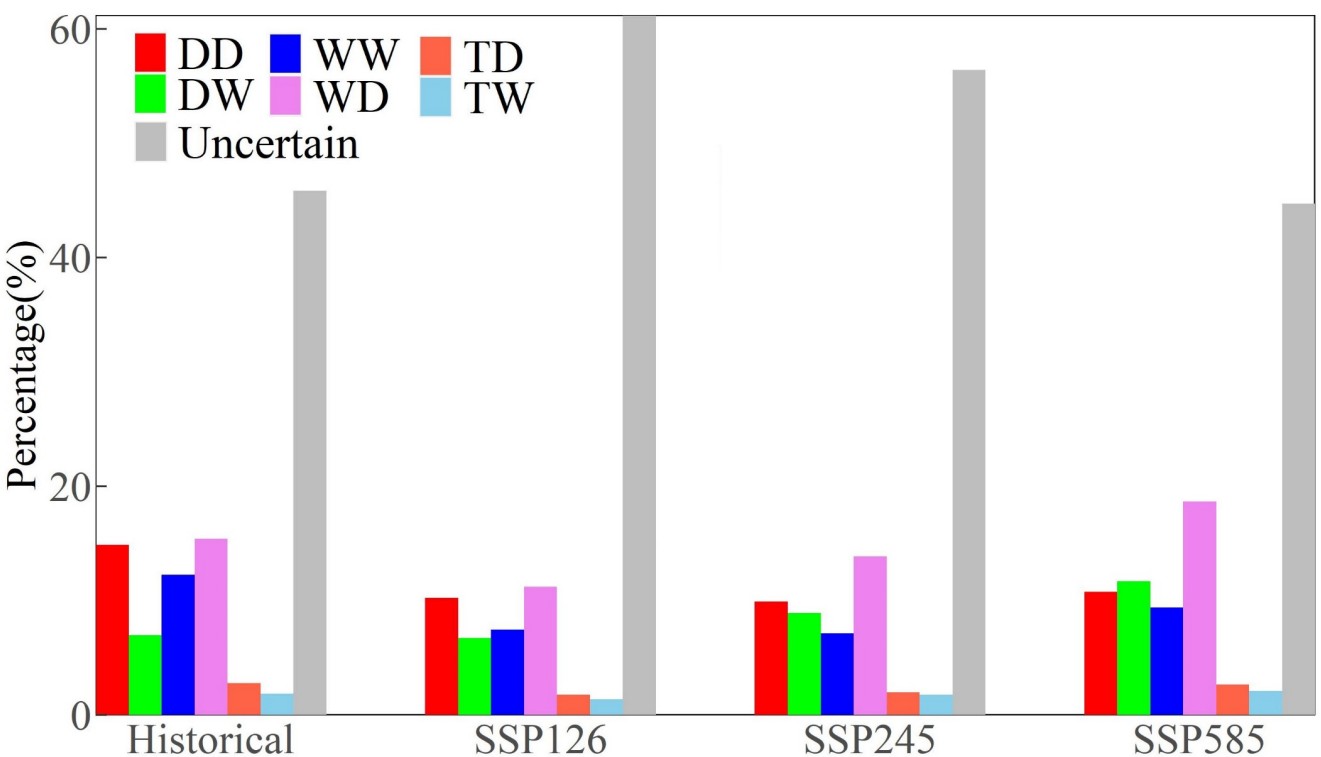





**Figure 5: Global statistics of the regions with various patterns during the historical (1985-2014) and future (2071-2100) period under SSP126, SSP245, and SSP585 scenarios. Note: The DD indicates the dry gets drier; the DW indicates the dry gets wetter; the WW indicates the wet gets wetter; the WD indicates the wet gets drier; the TD indicates the transition gets drier; the TW indicates the transition gets wetter; the Uncertain indicates the regions showing insignificant trends in TWS-DSI.**

## 4 Conclusion

Our reexamination for the DDWW paradigm from the perspective of TWSA using a large ensemble of GRACE reconstructions, global hydrological models, land surface models, global climate models during the historical (1985-2014) and future (2071-2100) period yields the following conclusions. First, during the historical period, 31.1% and 22.2% of grid cells present significant drying and wetting trend, respectively ($p<0.05$). During the future period under climate change, the proportion of drying area with a significant slope increases from SSP126 (23.3%) to SSP585 (32.3%) scenario. A similar growth is detected in the percentage with significant wetting trends, which reaches 15.6%, 17.8%, and 23.3% under SSP126, SSP245, and SSP585 scenario, respectively. Second, 27.1% of the area shows the DDWW paradigm, in which 14.9% and 12.2% of area is drying and wetting, respectively during the period 1985-2014. 22.4% of the area, however, show the opposite pattern like "dry gets wetter" (DW, 7%) and "wet gets drier" (WD, 15.4%), respectively. Under climate change, the proportion of area supporting the DDWW paradigm is 17.7%, 17.0%, and 20.0% under SSP126, SSP245, and SSP585 scenarios, respectively. Alternatively, the area opposing the DDWW paradigm achieves 17.9%, 22.7%, and 30.3%, respectively. Sensitivity analysis on different significance levels and choices of datasets gives more evidence for the reliability of our reexamination. Our insights from the TWSA perspective highlight that the widely-used DDWW paradigm is still challenging in both historical and future periods under climate change.

**Data Availability**

The data used in this study are as follows: GRACE (CSR, http://www2.csr.utexas.edu/grace/; JPL, http://podaac.jpl.nasa.gov/grace; GSFC, https://earth.gsfc.nasa.gov/geo/data/grace-mascons), GRACE reconstructions (CSR, https://doi.org/10.1029/2021GL093492; JPL and GSFC, https://doi.org/10.5194/essd-11-1153-2019), GHMs (WGHM, https://gmd.copernicus.org/articles/14/1037/2021/; VIC, https://ldas.gsfc.nasa.gov/gldas; PCR-GLOBWB, https://globalhydrology.nl/research/models/pcr-globwb-2-0/), LSMs (Noah, https://ldas.gsfc.nasa.gov/gldas; CPC, https://www.psl.noaa.gov/data/gridded/data.cpcsoil.html; CLSM, https://ldas.gsfc.nasa.gov/gldas), GCMs (https://esgf-node.llnl.gov/projects/cmip6/).

**Supplement**

The supplement related to this article is available online



## Author contributions

Jinghua Xiong conceived and designed the experiments. Jinghua Xiong performed the experiments. Jinghua Xiong analyzed the data. Jinghua Xiong, Shenglian Guo, Jie Chen, and Jiabo Yin wrote and edited the paper

## Competing interests

The authors declare that they have no conflict of interest.

## Acknowledgments

This study was financially supported by the National Key Research and Development Program of China (2021YFC3200303), the National Natural Science Foundation of China [U20A20317]. The numerical calculations in this paper have been done on the supercomputing system in the Supercomputing Center of Wuhan University.

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
