# Peer review of "A reexamination of the dry gets drier and wet gets wetter paradigm over global land: insight from terrestrial water storage changes"

_Hydrology and Earth System Sciences, 2021_

## Author Comment (AC1)

**Legend**

Reviewers' comments

Authors' responses

Direct quotes from the revised manuscript

We thank the reviewer for his/her time in reading our manuscript and detailed comments on our manuscript. Point-by-point replies to the comments or suggestions made can be found below.

**Reviewer #1:** This study re-examines the "dry gets drier and wet gets wetter" (DDWW) paradigm from the perspective of terrestrial water storage anomaly (TWSA) using a large ensemble of GRACE reconstructions, global hydrological models, and land surface modes. Based on the proportional percentages of different patterns, the results showed the consistent/opposite pattern with the DDWW and then the authors claimed that the paradigm faces challenge in both history (1985-2014) and future (2071-2100).

The topic is interesting and this study potentially provide a new perspective. However, I do not see the methods are convincing and the results are robust. First of all, I want to say that the dryness/wetness change itself contains different models, so it is not surprising to find the change models that do not follow the "DDWW" paradigm on a global scale.

Response: We thank the reviewer for recognizing the potential of the manuscript's new perspective and his/her detailed suggestions for improvement. All the concerns raised have been addressed in the revised manuscript. We hope the modified text along with the supplementary analyses and discussions will put forward the results in a much more robust way.

**Major comments:**

(1) My largest concerns are: Can GRACE observed TWS be used to estimate land surface dryness/wetness trends? How well (sensitive) can TWSA represent long-term trends in dryness/wetness across land surfaces? Is it better than traditional drought indices (e.g., the SPEI, PDSI or other methods)? There is no authoritative study demonstrating the suitability and applicability of the GRACE observed TWS in capturing surface dryness/wetness trends, especially on a global scale. Please note that, generally, the GRACE observed TWSA is applied to monitor changes in groundwater, land-ice evolution, and drought/flood events which occur on a short-term scale (see References). Hydrological processes are complex, but indices are often based on a relatively simple calculation. I take an example to show my understanding here. In glacier-covered mountains, as the climate warms, ice/glaciers are degrading with an increase in runoff/soil moisture (moisten the land surface). Meanwhile, as the mass decreases (water flows away), what GRACE observes is a decrease trend in gravity (drying). TWSA estimated trend and the real surface dry/wet trend can be absolutely opposite. Thus, changes in TWS do not equal to changes in surface dryness/wetness. Right?

Let's continue this topic and look at the Figure 3a. Over the past few decades, glacier melting and increasing runoff/wet trend in the southwest of Tibetan Plateau have been reported (e.g., the Fig. 4b of Yang et al., 2019), but the TWSA detect a drying trend in historic period. This clearly shows that the use of TWSA to estimate surface dry/wet trend is not robust. In addition, terrestrial water storage anomaly contains the information of changes in groundwater. With increasing human activity, large-scale pumping reduces groundwater (i.e., TWS observed a decrease trend) whereas the groundwater pumping and agricultural irrigation can moisten the land surface. I feel that the subtle effects of pumping and agricultural irrigation on dryness/wetness changes also cannot be captured by the TWSA. Therefore, I cannot confirm how valuable the perspective proposed by the author is for the capture of surface dry/wet changes.

Response: We thank the reviewer for the enlightening and detailed comment. We would like to discuss several aspects outlined in detail below.

Firstly, it is worth noting that this study focuses on the status of the terrestrial water storage instead of just the land surface, which is jointly affected by the soil moisture, groundwater, snow and ice, river and lake, and water stored in the vegetation (Rodell et al., 2018). Therefore, as important indicators of the terrestrial system, TWSA and TWSA-inferred measures including the total storage deficit index (TDSI) (Nie et al., 2018), GRACE-drought severity index (GRACE-DSI) (Zhao et al. 2017), and Water Storage Deficit Index (WSDI) (Thomas et al., 2014) have been widely used in the assessment of the trends in dryness and wetness globally (van Dijk et al., 2014; Xie et al., 2016; Xie et al., 2019). Compared with previous measures that focus on the hydrometeorological fluxes (e.g., SPI, SPEI, and PDSI) or the single component of the land system like soil moisture (SSI), groundwater (SGI), and runoff (SRI), our currently employed TWSA-based metric (i.e., TWS-DSI) can offer an alternative perspective to assess the global dryness and wetness of the land systems. We completely agree with the reviewer that the indices are generally based on a simplified calculation, and the changes in TWS do not equal changes in surface dryness/wetness due to the movement of water to and from the aquifer systems (e.g., surface water groundwater interaction, root zone moisture distribution, deep percolation, etc.). However, what our study concentrates on is the overall status of the land system, instead of either of the single part (e.g., surface water), which is the vertical integration of all the components of the terrestrial system.

Secondly, we would like to explain and emphasize the performance disparity and added value of our results by taking two regions as the example cases (see Figure R1). Figure R2 presents the monthly time series of GRACE TWSA from the CSR mascon solution and its main components from the WGHM in the southwest of the Tibetan Plateau (i.e., upper Brahmaputra River basin) during the period 2003-2016. It can be seen that both GRACE TWSA and WGHM modelling results present significant ($p<0.05$) decreasing trends with rates of $-18.50\pm1.14$ (GRACE) and $-2.42\pm0.33$ mm/a, respectively (see Table R1). Meanwhile, the snow water also shows a significant downward trend at a rate $-2.34\pm0.27$ mm/a due to increasing air temperature (Meng et al., 2019), but there is an upward trend ($0.34\pm0.09$ mm/a) of surface water storage during the same period, which might be due to the redistribution of water from ice/glaciers/snow to the land surface and increasing precipitation (Chun et al., 2020). Another example in northwest India also reflects a similar principle. This region, consisting of three states (Haryana, Punjab, and Rajasthan), has experienced one of the severest groundwater depletion in the world over decades due to water pumping for irrigation (Rodell et al., 2009; Dangar et al., 2021). Both GRACE and WGHM reveal the significant decreasing trend of TWSA, with slopes of $-17.65\pm2.55$ and $-30.82\pm1.42$ mm/a, respectively (Figure R2). The significant negative trend of GWSA is as low as $-31.99\pm1.27$ mm/a. Nevertheless, the regional surface water (and even soil moisture) shows a slight increasing slope of $0.90\pm0.25$ mm/a ($p<0.05$), which might be triggered by the moistening of explored groundwater to the land surface by human interventions, as rightly mentioned by the reviewer, and depleting net

precipitation (Chen et al., 2014). Generally, both the upper Brahmaputra and northwest India have been considered as drying regions from the perspective of TWSA, though the opposite results will be obtained if only the surface water (and soil moisture) is selected for dryness/wetness assessment. Having discussed that, given the consistency of the TWSA decreasing/increasing trends with the previous studies, we believe that the presented results are robust.

We regret the error and consequential confusion between the land surface dryness/wetness and the integrated land system dryness/wetness. In the revised version, we have rectified the same and stressed our research objectives of examining the dry gets drier and wet gets wetter paradigm over global land from the TWSA perspective. Our findings can provide novel implications for the large-scale detection of dryness and wetness of terrestrial systems in a changing environment.

[Figure]

Figure R1. The location and the elevation of the two study cases, i.e., (a) upper Brahmaputra and (b) northwest India.

[Figure]

Figure R2 Time series of TWSA and its main components from the GRACE and WGHM during 2003-2016 in the (a) upper Brahmaputra River basin and (b) northwest India. Note: The dashed and the solid black lines represent the GRACE and WGHM data, respectively. TWSA is the terrestrial water storage anomaly, RSA is the river storage anomaly, SWSA is the surface water storage anomaly, SWEA is the snow water equivalent anomaly, CWSA is the canopy water storage anomaly, SMSA is the soil moisture anomaly, GWSA is the groundwater storage anomaly. Anomalies in all the components are corresponding to the baseline period of 2004-2009. Scales of the left and right y-axis are set differently for clarity and better comprehension.

Table R1 Linear trends in TWSA and its main components from the GRACE and WGHM during 2003-2016 in the upper Brahmaputra and northwest India. Please refer to Figure R2 for abbreviations of different water storage components. The significant (p<0.05) trends are shown in bold fonts.

| Region | Upper Brahmaputra | | Northwest India | |
|---|---|---|---|---|
| Variable | Slope (mm/a) | Error (mm/a) | Slope (mm/a) | Error (mm/a) |
| GRACE TWSA | **-18.50** | 1.14 | **-17.65** | 2.55 |
| WGHM TWSA | **-2.42** | 0.33 | **-30.82** | 1.43 |
| WGHM RSA | 0.08 | 0.08 | **0.12** | 0.03 |
| WGHM SWSA | **0.34** | 0.09 | **0.90** | 0.25 |
| WGHM SWEA | **-2.34** | 0.27 | 0.00 | 0.00 |
| WGHM CWSA | 0.00 | 0.00 | 0.00 | 0.00 |
| WGHM SMSA | -0.08 | 0.04 | 0.14 | 0.12 |
| WGHM GWSA | **-0.42** | 0.07 | **-31.99** | 1.27 |

**References:**

Chen, J., Li, J., Zhang, Z., Ni, S. 2014. Long-term groundwater variations in Northwest India from satellite gravity measurements Glob. Planet. Change, 116, pp. 130-138, https://doi.org/10.1016/j.gloplacha.2014.02.007

Chun, K.P., He, Q., Fok, H.S., Ghosh, S., Yetemen, O., Chen, Q., Mijic, A. 2020. Gravimetry-based water storage shifting over the China-India border area controlled by regional climate variability. Sci. Total. Environ. 714, 136360. https://doi.org/10.1016/j.scitotenv.2019.136360

Dangar, S., Asoka, A., Mishra, V. 2021. Causes and implications of groundwater depletion in India: a review. J. Hydrol., 596 (2021), Article 126103, 10.1016/j.jhydrol.2021.126103

Meng, F., Su, F., Li, Y., Tong, K. 2019. Changes in terrestrial water storage during 2003–2014 and possible causes in Tibetan Plateau. Journal of Geophysical Research: Atmospheres, 124, 2909–2931. https://doi.org/10.1029/2018JD029552

Nie, N., W. Zhang, H. Chen, and H. Guo, 2018: A Global Hydrological Drought Index Dataset Based on Gravity Recovery and Climate Experiment (GRACE) Data. Water Resour. Manag., 32, 1275–1290. https://doi.org/10.1007/s11269-017-1869-1

Rodell, M., Velicogna, I., Famiglietti, J.S. 2009. Satellite-based estimates of groundwater depletion in India. Nature, 460, pp. 999-1002, https://doi.org/10.1038/nature08238

Rodell, M., J. S. Famiglietti, D. N. Wiese, J. T. Reager, H. K. Beaudoing, F. W. Landerer, and M. H. Lo, 2018: Emerging trends in global freshwater availability. Nature, 557, 651–659, https://doi.org/10.1038/s41586-018-0123-1.

Thomas, A. C., J. T. Reager, J. S. Famiglietti, and M. Rodell, 2014: A GRACE- based water storage deficit approach for hydrological drought characterization. Geophys. Res. Lett., 41, 1537–1545, https://doi.org/10.1002/2014GL059323.

Van Dijk, A.I.J.M., Renzullo, L.J., Wada, Y., Tregoning, P. 2014. A global water cycle reanalysis (2003–2012) merging satellite gravimetry and altimetry observations with a hydrological multi-model ensemble Hydrol. Earth Syst. Sci., 18 (8), p. 2955.

Xie, Z., Huete, A., Cleverly, J., Phin, S., McDonald-Madden, E., Cao, Y., Qin, F. 2019. Multi-climate mode interactions drive hydrological and vegetation responses to hydroclimatic extremes in Australia. Remote Sens. Environ. 231: 111270. https://doi.org/10.1016/j.rse.2019.111270

Xie, Z., Huete, A., Restrepo-Coupe, N., Ma, X., Devadas, R., Caprarelli, G. 2016. Spatial partitioning and temporal evolution of Australia's total water storage under extreme hydroclimatic impacts, Remote Sens. Environ., 183, 43–52, https://doi.org/10.1016/j.rse.2016.05.017

Zhao, M., A. Geruo, I. Velicogna, and J. S. Kimball, 2017: Satellite observations of regional drought severity in the continental United States using GRACE-based terrestrial water storage changes. J. Clim., 30, 6297–6308, https://doi.org/10.1175/JCLI-D-16-998 0458.1.

(2) My another question is why the authors confirm that an ensemble way is more reliable than a single way? This draft does not show the individual results of different methods, nor does it compare the differences in these results, so I can't be sure that the way of ensemble is reliable. In the Figure

S2, there are gaps between gravity satellite observations and climate model simulations. Besides, why the authors use the GRACE observation to correct the CMIP6 historical simulation? Do you think the simulation of CMIP6 is unreliable (relative to GRACE)? Why? How to define and calculate the TWS in hydrological models, CMIP6, and land surface models? Are they talking about the same thing (and same with the GRACE's TWS)? How different are the estimated TWS between these methods? I don't think simply integrating the various outputs is a right path, because of the inherent scale differences between climate models, hydrological models, and satellite observations, and I think the TWS in these methods is not the same object.

Response: We thank the reviewer for the comment. Please find the detailed explanation of all the concerns below.

*The rationale of selecting multi-model ensemble:* We have compared the GRACE observations and each individual simulation, as well as their ensemble mean, during April 2002-December 2014. The global distributions of NRMSE are shown in Figure R3. Three GRACE reconstructions present relatively lower error than the global hydrological models and land surface models, especially in the high-latitude northern hemisphere where snow, ice, and glaciers contribute more to TWS than other regions, which is not considered in most of the global models. The ensemble-mean solution illustrates the reasonably good accuracy with the NRMSE generally below 0.2, highlighting the reduced uncertainty compared with individual solutions. It is not surprising that the GRACE reconstructions compare better than other data because they are directly calibrated with the GRACE measurements during 2002-2017. While their performances need more validation beyond the GRACE era (i.e., prior to April 2002 and during July 2017-June 2018). Similar patterns are also discovered from the probability density functions of NRMSE, of which there is an overall negative deviation in the ensemble-mean relative to other solutions except for the CSR reconstruction (see Figure R4). The Taylor diagram also confirms the enhanced accuracy of TWSA after taking the average of the large ensemble, with the increased correlation and decreased standard deviation.

In addition, the comparison between CMIP6-inferred and GRACE TWSA in the past (April 2002-December 2014) is conducted (see Figure R5). The spatial distributions clearly show that the ensemble mean of eight global climate models outperforms each member globally, particularly in some parts of Australia, southern Africa, and North America. An overall decrease in NRMSE is observed according to the probability density functions, which is also detected from the Taylor diagram results (see Figure R6). Generally, the ensemble way could reduce the uncertainty of individual data sources in the past and future period, and the GRACE-based bias correction method also reduces the bias in projected TWSA. We have added discussion for the multi-source uncertainty from different data sources and methods in the revised manuscript.

*Better representation of TWS in GRACE:* Unlike GRACE observations other historical simulations, e.g., the CMIP6 outputs, are generally affected by large systematic biases owning to the uncertainty in the projected meteorological forcing and the lack of complete parameterizations for surface water, canopy water, and groundwater storage, even some of the key climate inputs such as precipitation

and air temperature have been proven underestimated or overestimated nearly over the global land (Kim et al., 2020; Wu et al., 2021). Therefore, we applied the trend-preserving method to perform the bias correction for the CMIP6 TWSA in conjunction with the GRACE observations. The detailed calculation procedures have been introduced in a recent study (Xiong et al., 2022).

We acknowledge that there exist differences between different sources of TWS including the GRACE, GHMs, LSMs, and global climate models (GCMs) from CMIP6 (see Table R2). Many LSMs only simulate snow and soil moisture compartments, whereas most GHMs simulate all storage compartments, excluding glaciers. For example, the VIC and CPC models lack the modules of groundwater and surface water storage, which could affect the TWSA over regions with intensive human intervention such as groundwater abstraction and reservoir operation. The lack of simulation in water stored in lakes and reservoirs, aquifers, and vegetation inevitably make multiple GCMs suffer from large uncertainty compared with the GRACE observations that represent the full signal of TWSA components, although the raw GRACE data also experiences signal leakage and spatial resample on the grid-scale (Scanlon et al., 2016). In addition, the associated GRACE reconstructions based on statistical methods and machine learning techniques may also overestimate or underestimate the true TWSA due to missing the inherent physics of the hydrological processes (Li et al., 2021). However, given the fact that no single model, satellite solution, and/or reconstruction perform best everywhere globally, it is still evident that using the ensemble mean of different model outputs can help in eliminating the systematic and/or bias errors implicit to the individual outputs, at least on a global scale (Long et al., 2017; Scanlon et al., 2018; Sun et al., 2021).

*Results from the individual model outputs:* Apart from the ensemble mean results in Figure R7 that 28.1% (23.3%) of global land confirms (opposes) the DDWW paradigm in the past and the percentage supporting the DDWW pattern is lower than 20% in the future, we also carried out an independent analysis at the individual member level (see Figure R8). For the historical period, a clear overestimation of the CSR reconstructions is detected with 42.4% of the area agreeing with the DDWW pattern, and 36.6% showing the opposite situation. Moreover, the modelled results from VIC and WGHM illustrate the underestimation of the area validating the DDWW paradigm, reaching 15.6% (WGHM) and 12.2% (VIC), respectively. Their proportion with the opposite DDWW paradigm is 10.2% (WGHM) and 17.8% (VIC), respectively. Therefore, it can be concluded that the differences among different members of the ensemble limitedly affect the evaluation of the DDWW during the historical period. In the future, the GFDL-ESM4 model presents overestimation but the IPSL-CM6A and CanESM5 models have underestimation for different percentages compared with the ensemble-mean. Specifically, the area dominated by the DDWW paradigm changes from 8.9% (CanESM5) to 21.9% (GFDL-ESM4), while that showing the opposite pattern ranges from 7.8% (CanESM5) to 14.8% (GFDL-ESM4) under the SSP126 scenario. For the SSP245 scenario, the DDWW-validated regions account from 7.4% (CanESM5) to 21.5% (GFDL-ESM4), the opposite pattern occurs over a range from 9.7% (CanESM5) to 16.0%

(GFDL-ESM4) of the global land. The proportion supporting the DDWW paradigm varies from 10.4% (CanESM5) to 24.0% (GFDL-ESM4), while that presenting the opposite pattern ranges from 8.4% (CanESM5) to 22.3% (GFDL-ESM4) under the SSP585 scenario. Overall, the comparatively large difference among various models might source from unforced internal climate variability of distinctive CMIP6 members and the different emission scenarios (Kumar et al., 2015).

In the revised version of the manuscript, we have clarified the reasons why we use GRACE to perform bias-correction for GCMs data, presented the individual results for the examination of the DDWW paradigm, and added discussion on the uncertainty sourced from different models, satellite products, and methods. The differences in the TWSA from distinctive sources are also discussed.

[Figure]

Figure R3. Global distribution of NRMSE between TWSA derived from the GRACE mission and each member and the ensemble-mean of DATASET from Apr. 2002 to Dec. 2014.

[Figure]

Figure R4. (a) Probability density function and (b) Taylor diagram of NRMSE between TWSA derived from the GRACE mission and each member and the ensemble-mean of DATASET from Apr. 2002 to Dec. 2014.

[Figure]

Figure R5. Global distribution of NRMSE between TWSA derived from the GRACE mission and each member and the ensemble-mean of the eight GCMs from Apr. 2002 to Dec. 2014.

[Figure]

Figure R6. (a) Probability density function and (b) Taylor diagram of NRMSE between TWSA derived from the GRACE mission and each member and the ensemble-mean of eight GCMs from Apr. 2002 to Dec. 2014.

Table R2. Summary of attributes of different models used in this study

| Dataset | GRACE | WGHM | VIC | PCR-GLOBWB | Noah | CPC | CLSM | CMIP6 |
|---|---|---|---|---|---|---|---|---|
| Parameter | Satellite | | GHM | | | LSM | | GCM |
| Surface water storage | √ | √ | × | √ | × | × | × | × |
| Soil moisture | √ | √ | √ | √ | √ | √ | √ | √ |
| Groundwater storage | √ | √ | × | √ | × | × | √ | × |
| Canopy water | √ | √ | × | √ | √ | × | √ | × |
| Snow water | √ | √ | × | √ | √ | × | √ | √ |
| Soil layers (no.) | / | 1 | 3 | 2 | 4 | 10 | 10 | 5~10 |
| Soil depth (m) | / | 2 | 2 | 1.5 | 2 | 1.6 | 1 | 2~10 |

[Figure]

Figure R7 Global statistics of the regions with different patterns during the historical (1985-2014) and future (2071-2100) periods under SSP126, SSP245, and SSP585 scenarios. Note: DD indicates the dry gets drier; DW indicates the dry gets wetter; WW indicates the wet gets wetter; WD indicates the wet gets drier; TD indicates the transition gets drier; TW indicates the transition gets wetter; Uncertain indicates the regions showing insignificant (p>0.05) trends in TWS-DSI.

[Figure]

Figure R8. Same as Figure R7, but based on individual dataset during the (a) historical (1985-2014) and future (2071-2100) periods under (b) SSP126, (c) SSP245, and (d) SSP585 scenarios, respectively.

**References:**

Kim, Y.H., Min, S.K., Zhang, X., Sillmann, J., Sandstad, M. 2020. Evaluation of the CMIP6 multi-model ensemble for climate extreme indices. Weather Clim. Extremes, 29, p. 100269, https://doi.org/10.1016/j.wace:2020.100269

Kumar, S., Allan, R.P., Zwiers, F., Lawrence, D.M., Dirmeyer, P.A., 2015. Revisiting trends in wetness and dryness in the presence of internal climate variability and water limitations over land. Geophys. Res. Lett. 42, 10867–10875. https://doi.org/10.1002/2015GL066858

Li, F., Kusche, J., Chao, N., Wang, Z., Loecher, A., 2021. Long-Term (1979-Present) Total Water Storage Anomalies Over the Global Land Derived by Reconstructing GRACE Data. Geophys. Res. Lett. 48, e2021GL093492. https://doi.org/10.1029/2021GL093492

Long, D., Pan, Y., Zhou, J., Chen, Y., Hou, X., Hong, Y., Scanlon, B. R., Longuevergne, L. 2017. Global analysis of spatiotemporal variability in merged total water storage changes using multiple GRACE products and global hydrological models, Remote Sens. Environ., 192, 198–216, https://doi.org/10.1016/j.rse.2017.02.011

Scanlon, B. R., Zhang, Z., Save, H., Wiese, D. N., Landerer, F. W., Long, D., Longuevergne, L., Chen, J. 2016. Global evaluation of new GRACE mascon products for hydrological applications. Water Resour. Res., 52, 9412–9429.

Scanlon, B.R., Zhang, Z., Save, H., Sun, A.Y., Müller Schmied, H., van Beek, L.P.H., Wiese, D.N., Wada, Y., Long, D., Reedy, R.C., 2018. Global models underestimate large decadal declining and rising water storage trends relative to GRACE satellite data. Proc. Natl. Acad. Sci. U. S. A. 201704665.

Sun, A.Y., Scanlon, B.R., Save, H., Rateb, A., 2021. Reconstruction of GRACE total water storage through automated machine learning. Water Resour. Res., 57(2). 787 DOI:10.1029/2020WR028666

Wu, R.-J., Lo, M.-H., Scanlon, B.R., 2021. The annual cycle of terrestrial water storage anomalies

in CMIP6 models evaluated against GRACE data. J. Clim. 34, 8205–8217. https://doi.org/10.1175/JCLI-D-21-0021.1

Xiong, J., Guo, S., Yin, J., Ning, Z., Zeng, Z., Wang, R. 2022. Projected changes in terrestrial water storage and associated flood potential across the Yangtze River basin. Sci. Total Environ. 817, 152998. https://doi.org/10.1016/j.scitotenv.2022.152998

(3) I found a fault in the fundamental calculation. The presented area percentages are calculated by the number of grids, which are not the real area of the Earth sphere. Such calculation can greatly reduce the proportion in the tropics, but we think the "wet wetter" paradigm is generally well followed there.

Response: We thank the reviewer for pointing out this fault. We have performed the re-calculation based on the actual area instead of the number of grid cells to evaluate the DDWW paradigm over global land (see Figure R9). All the results have been updated in the revised manuscript.

[Figure]

Figure R9. Spatial distribution of actual area of the 1° grid cells over global land.

(4) The titles of section 3.1 and section 3.2 are the same, i.e., "Global trends of dryness and wetness". How rough! Despite an admirable effort by the authors to process data and conduct calculations, the manuscript lacks discussion and more is showing calculation results. Uncertainties regarding to the new methods and results should be fully discussed.

Response: We apologize for this oversight. The sub-title of section 3.2 has been changed to "Assessment of the DDWW Paradigm". As suggested, we have added discussions for the uncertainty regarding the new methods and data in the revised manuscript. Please see the section "3.3 Uncertainties and Implications" for more details. Please find the appended discussion below:

Each ensemble member of the DATASET has embedded uncertainties inherently originating from one or more of forcing variables, simplified assumptions of complex processes in the models and their physical structure, retrieval algorithms, and systematic biases, which inevitably have propagated to the results presented herein. For example, the original GRACE mascon observations contain the measurement error and signal leakage at gridded scale, and they further spread into the reconstruction of TWSA when training via the statistical methods (Humphrey and Gudmundsson,

2019; Li et al., 2021a). Unlike observed GRACE and reconstructed GRACE-like data, the TWSA simulations from the GHMs, LSMs, and GCMs are featured by incomplete representation (Table S2). They are generally based on the simplified hydrological processes, resulting in the missing some of the TWSA components. For example, the widely used Noah model lacks the surface water and groundwater storage in TWSA, and all the GCMs can only simulate the snow water and soil moisture within a limited depth from 2 to 10 m below land surface (Xiong et al., 2021b; Wu et al., 2021). Moreover, the eight CMIP6 GCMs are forced with the future projections of many meteorological variables such as precipitation and air temperature, which have been reported showing underestimation or overestimation nearly over the global land (Eyring et al., 2016; Kim et al., 2020). Despite employing bias correction with GRACE data, propagated uncertainty from the forcing and models can influence the accuracy of TWSA simulations (Xiong et al., 2022). Although it is challenging to explicitly attribute and quantify these uncertainties in absence of a true reference observation dataset, the ensemble averaging method has been used to integrate the multi-source TWSA data. The global distributions of NRMSE between GRACE observations and each ensemble member and their mean during April 2002-December 2014 show improved performance of the latter (Figure S7). Three GRACE reconstructions present relatively lower error than the GHMs and LSMs, especially in the high-latitude northern hemisphere where snow, ice, and glaciers contribute more to TWS than other regions, which is not considered in most of the global models. The ensemble-mean solution illustrates the reasonably good accuracy with the NRMSE generally below 0.2, highlighting the reduced uncertainty compared with the individual solution. It is not surprising that the GRACE reconstructions compare better than other data because they are directly calibrated with the GRACE measurements during 2002-2017. While their performances need more validation beyond the GRACE era (i.e., prior to April 2002 and during July 2017-June 2018). Similar patterns are also evident from the probability density functions of NRMSE, of which there is an overall negative deviation in the ensemble-mean relative to other solutions except for the CSR reconstruction (see Figure S8). This outperformance of the ensemble dataset is ascertained by the increased correlation and decreased standard deviation as shown by the Taylor diagram (Figure S8). In addition, the comparison between GCM-modelled and GRACE TWSA in the past (April 2002-December 2014) is conducted (see Figure S9). The spatial distributions clearly show that the ensemble-mean of eight GCMs outperforms each member globally, particularly in Australia, southern Africa, and North America. An overall decrease in NRMSE is observed according to the probability density functions, which is also detected from the Taylor diagram results (see Figure S10).

Further, we carried out an independent analysis at the individual member level (see Figure S11). For the historical period, a clear overestimation of the CSR reconstructions is detected with 42.4% of the area agreeing with the DDWW pattern, and 36.6% showing the opposite situation. Moreover, the modelled results from VIC and WGHM illustrate the underestimation of the area validating the DDWW paradigm, reaching 15.6% (WGHM) and 12.2% (VIC), respectively. Their proportion with

the opposite DDWW paradigm is 10.2% (WGHM) and 17.8% (VIC), respectively. Therefore, it can be concluded that the differences among different members of DATASET limitedly affect the evaluation of the DDWW during the historical period. In the future, the GFDL-ESM4 model presents overestimation but the IPSL-CM6A and CanESM5 models have underestimation for different percentages compared with the ensemble mean. Specifically, the area dominated by the DDWW paradigm changes from 8.9% (CanESM5) to 21.9% (GFDL-ESM4), while that showing the opposite pattern ranges from 7.8% (CanESM5) to 14.8% (GFDL-ESM4) under the SSP126 scenario. For the SSP245 scenario, the DDWW-validated regions account from 7.4% (CanESM5) to 21.5% (GFDL-ESM4), the opposite pattern occurs over a range from 9.7% (CanESM5) to 16.0% (GFDL-ESM4) of land. The proportion supporting the DDWW paradigm varies from 10.4% (CanESM5) to 24.0% (GFDL-ESM4), while that presenting the opposite pattern ranges from 8.4% (CanESM5) to 22.3% (GFDL-ESM4) under the SSP585 scenario. Overall, the comparatively large difference among various models might source from unforced internal climate variability of distinctive CMIP6 members and different emission scenarios (Kumar et al., 2015).

Our choice of the significance level (i.e., 0.05) may also affect the rationale of the DDWW examination results, thus different significance levels are alternatively tested (see Figure S11). At a significance level of 0.01, 22.2% of land area agrees well with the DDWW theory, while the 17.1% of area illustrates the opposite pattern during the period 1985-2014. As for the 0.1 significance level, the DDWW-validated regions account for 30.6% of the total area, with 25.4% of land agreeing with the opposite hypothesis. In the future period, a similar pattern is discovered that both DDWW-confirmed and DDWW-opposed regions are increasing on account of the enhancement of projected strength of radiative forcing, with the reduction of the area showing insignificant trends in wetting and drying. However, the magnitudes of results at the 0.01 significance level are generally lower than that at the 0.1 significance level due to the different thresholds of detected trends in drying and wetting.

Despite the multisource uncertainties, our study can provide important implications for the long-term trends in dryness/wetness over global land in the past and future from the perspective of TWSA. Compared with other widely used indexes that are purely derived from the hydrometeorological variables (e.g., SPI, SPEI, and PDSI) or incorporate a single component of the TWSA (e.g., SSI, SGI, and SRI), our developed TWS-DSI describes the overall status of the land system, which is jointly influenced by different components including soil moisture, river runoff, and groundwater that play different roles in the hydrological cycle (Tapley et al., 2019). The new insights benefit the comprehensive evaluation of terrestrial conditions over regions where some parts of TWSA (e.g., groundwater storage and snow water) have been rapidly depleting due to intensive human activities and warming climate worldwide, including the Qinghai-Tibet Plateau and northwest India (Rodell et al., 2009; Xing et al., 2021). Furthermore, the projected changes in global TWSA and associated TWS-DSI improve our understanding of the large-scale hydrological response under climate change, particularly in regions with strong human interventions such as the

south and east of Asia. Despite the magnitude bias from satellite products, simulations of LSMs and GHMs, and GCMs projections, the ensemble averaging method has presented an effective and efficient ability to alleviate the multi-source uncertainty, which can be further applied over data-sparse areas globally with limited in-situ observations like Africa and central Asia. In addition, the regional aggregation of the analysis based on the IPCC AR6 SREX references regions can supply valuable inferences for policymakers and stakeholders for better water resources management in a changing environment (Iturbide et al., 2020).

**Specific comments:**

(1) Line 13-14: Why the sum of the patterns is 27.1% plus 22.4% (not 100%)? What about other patterns?

Response: We thank you for the informative comment. There is a total of six patterns been detected including "the dry gets drier (DD, 16.7%)", "the dry gets wetter (DW, 8.4%)", "the wet gets wetter (WW, 11.4%)", "the wet gets drier (WD, 14.9%)", "the transition gets drier (TD, 2.6%)", and "the transition gets wetter (TW, 1.7%)". Apart from these patterns, there is 44.3% of the total area showing uncertain trends. Given our main objective of examining whether the DDWW paradigm holds true from the perspective of TWSA, the percentages of areas showing alternative patterns (i.e., DW, WD, TD, and TW) are not included in the Abstract section. However, we have revised and systematically summarized the examination results in the Conclusion section.

(2) Line 20: What's the meaning of "fresh availability"?

Response: We regret this error. We have changed it with "freshwater availability" in the revised manuscript.

(3) Line 25: What do you mean "enhance"? What do you mean "vice versa"?

Response: We are sorry for the confusion. We have clarified this statement in the revised manuscript as follows:
This deficit is expected to increase due to the enhancement of atmospheric water vapor in humid regions (i.e., convergence zones) under a warming climate, and decrease over arid regions (i.e., divergence zones) (Durack et al., 2012).

(4) Line 26-27: "in hydrologic cycle under climate change in both regional and global scales". Is this expression a bit exaggerated?

Response: As suggested, we have weakened and re-organized this sentence as follows:
The DDWW paradigm has been used to represent the historical and future trends in various constituent components of the hydrologic cycle on regional (Chou et al., 2009; Allan et al., 2010;

Hu et al., 2019; Zeng et al., 2019) and global scales (Held and Soden, 2006; Donat et al., 2016).

(5) Line 29-30: "rational"-->"rationale". Do so many references really question the rationale of DDWW?

Response: Thank you for pointing this out. We have corrected the references list. The revised sentence reads as below:

However, the rationale of DDWW mechanism is recently questioned at different levels through the growing accessibility of datasets, models, and indicators (Polson and Hegerl, 2017; Yang et al., 2019; Y. Li et al., 2021).

(6) Line 40-41: "The uncertainties within previous studies are mainly sourced from different choices of measures and datasets". However, this study do not reduce such uncertainties, and there are also great uncertainties, as there are various data sources and interpolation methods.

Response: We have modified this sentence in the revised manuscript. To avoid the uncertainty sourced from different data sources, we developed an ensemble-mean method for the historical TWSA simulation and applied the GRACE observations to perform the bias correction for a large set of future projections of TWSA. The cross-comparisons within different data sources indicate that the ensemble-mean and the bias-corrected TWSA have better accuracy than the raw data, highlighting that the data uncertainty can be alleviated. Moreover, we have added discussion for the uncertainty derived from various datasets and interpolation methods in the revised manuscript. Please see our responses to Major comments 2 and 4 for more details.

(7) Line 45: It is true that "neglect the hydrological process on the land surface", but the TWSA used for estimating dryness/wetness is also an index and neglect the hydrological process.

Response: We thank you for this comment. Some widely used meteorological drought indices such as SPEI, SPI, and PDSI are undoubtedly intended to be convenient and generalized indicators of meteorological water deficit. However, their inconsistent ability to recreate hydrologically relevant patterns of the land system at regional scales owing to the trade-off between the simplicity of meteorological factors and the data needs as well as computational requirements of process-based variables, suggesting that they may not capture plot-specific terrestrial processes (e.g., preferential flow and groundwater recharge) (Barnard et al., 2021; Slette et al., 2020). Such hydrological processes can be considered and reflected by terrestrial-based indicators such as SSI (soil moisture), SGI (groundwater), SRI (runoff), and TWS-related measures. We have revised this statement in the revised manuscript as follows:

Meanwhile, some meteorological indices derived from precipitation and evapotranspiration like the standardized precipitation evapotranspiration index (SPEI), aridity index (AI), and standardized precipitation/evapotranspiration index (SPI/SETI) may not capture the integrated response of the

land system due to the trade-off between the simplicity of meteorological factors and the data needs as well as computational requirements of process-based variables (Huntington, 2006; Dai, 2011; Slette et al., 2020; Barnard et al., 2021).

**Reference:**

Barnard, D.M., Germino, M.J., Bradford, J.B., Connor, R.C., Andrews, C.M., Shriver, R.K. 2021. Are drought indices and climate data good indicators of ecologically relevant soil moisture dynamics in drylands? Ecol. indic. 133, 108379. https://doi.org/10.1016/j.ecolind.2021.108379

Slette, I.J., Smith, M.D., Knapp, A.K., Vicente-Serrano, S.M., Camarero, J.J., Beguería, S., 2020. Standardized metrics are key for assessing drought severity. Glob. Change Biol. 26, e1–e3.

(8) Line 47: "merely highlight differently single aspect of the water cycle, lacking the complete representation of the terrestrial water storage (TWS)". Why do you think a complete representation of TWS would be better than an index regarding single aspect of water cycle? I think there are already comprehensive drought/wet indices.

Response: Thank you for this suggestive comment. Given the difference in the formulation and subsequent implications, we do not directly compare the performances of the individual hydrometeorological or individual water storage component-based indices and our assessment. We acknowledge the significance of already existing such indices. However, our study provides a new perspective for the assessment of the DDWW paradigm over global land, which is a potentially crucial supplement to the current measures that just regarding the single aspect of the water cycle (e.g., SSI, SGI, and SRI). Therefore, we believe that the current TWS-based assessment put forward a measure of dryness/wetness/transition behaviours that evolve due to synergistic impacts of natural and anthropogenic drivers, which are exceedingly difficult to disentangle in the coupled human-natural systems (AghaKouchak et al., 2021; Rodell et al., 2018). The traditional drought/wet indices, though comprehensive in different contexts, do not represent such integrated variations. To better convey our message, we have modified the text as below:

A few indexes like the standardized soil moisture index (SSI), standardized groundwater index (SGI), and standardized runoff index (SRI), however, focus on a single aspect of the water cycle and do not describe the integrated status of the terrestrial water storage (TWS) (AghaKouchak, 2014; Wu et al., 2018; Guo et al., 2021). In the coupled human-natural systems, where the synergistic impacts of natural and anthropogenic drivers are exceedingly difficult to disentangle, an integrated representation of the land systems is of paramount importance for policymakers (AghaKouchak et al., 2021; Rodell et al., 2018).

**References**

AghaKouchak, A., Mirchi, A., Madani, K., Di Baldassarre, G., Nazemi, A., Alborzi, A. Anjileli, M., Azarderakhsh, H., Chiang, F., Hassanzadeh, E., Huning, L.S., Mallakpour, I., Martinez, A., Mazdiyasni, O., Moftakhari, H., Norouzi, H., Sadegh, M., Sadeqi, D., Van Loon, A.F., Wanders, N. 2021. Anthropogenic Drought: Definition. Rev. Geophys Challenges and Opportunities 10.1029/2019rg000683

Rodell, M., Famiglietti, J.S., Wiese, D.N., Reager, J.T., Beaudoing, H.K., Landerer, F.W., Lo, M.H. 2018. Emerging trends in global freshwater availability. Nature. 557(7707): 651–659.

10.1038/s41586-018-0123-1

(9) Line 50: "TWS consisting of water storage in surface water, soil moisture, groundwater, snow and ice, and canopies can physically provide integrated information..." But groundwater pumping reduces groundwater (TWS is decrease) and makes the surface wet.

Response: Thank you for the suggestion. We have modified this sentence. Please refer to our response to Major comment 1 for details.

(10) Line 93: What's the meaning of offline physically based?

Response: Thank you for the question. The physically-based VIC model was used to carry out the historical TWSA simulation during the period 1985-2014, which is not real time (offline) and forced with the Global Meteorological Forcing Dataset from Princeton University. We have revised the expression as follows:

The physically-based, semi-distributed, and grid-based VIC model is managed by the NASA Global Land Data Assimilation System Version 2.1 (GLDAS-v2.1) (Liang et al., 1994; Syed et al., 2008).

(11) Is it necessary to carry out regional studies according to the IPCC? The zoning studies make no sense in fact. They are just another display for the same results.

Response: Thank you very much for your comment. The IPCC SREX references regions have been popularly applied for the regional synthesis of historical trends and future climate change projections, particularly in the assessment of global dryness/wetness (e.g., the DDWW paradigm) (Yang et al., 2019; Balting et al., 2021; Dong et al., 2021). In this study, we not only downscaled the spatial distribution of trends in dryness/wetness, but also calculated the percentage of area with various patterns in different regions, to provide practical inference in managing the risks of extremes from the perspectives of policymakers and stakeholders. Moreover, given our use of eight global climate models from the CMIP6 archive, aggregating regional information with the recommended reference regions also benefits the climatic consistency and better representation of regional climate features as well as the representativeness of model results (Iturbide et al. 2020). Therefore, we would like to divide the global land area into 43 zones as defined by the IPCC Sixth Assessment Report (AR6) (IPCC, 2021). Moreover, we would like to kindly reiterate that, to the best of our knowledge, there has been no global study, focussing on examining the DDWW paradigm from a TWS perspective.

**References:**
Balting, D.F., AghaKouchak, A., Lohmann, G., Ionita, M. 2021. Northern Hemisphere drought risk in a warming climate. Clim. Atmos. Sci. 4(1), 61. https://doi.org/10.1038/s41612-021-00218-2.
Dong S, Sun Y, Li C, Zhang X, Min S-K and Kim Y-H 2021 Attribution of extreme precipitation with updated observations and CMIP6 simulations J. Clim. 34 871–81.
Yang, T., Ding, J., Liu, D., Wang, X., Wang, T., 2019. Combined use of multiple drought indices for

global assessment of dry gets drier and wet gets wetter paradigm. J. Clim. 32, 737–748. https://doi.org/10.1175/JCLI-D-18-0261.1

IPCC, 2021: Summary for Policymakers. In: Climate Change 2021: The physical science basis. Contribution of Working Group I to the Sixth Assessment Report of the Intergovernmental Panel on Climate Change [Masson-Delmotte, V., P. Zhai, A. Pirani, S. L. Connors, C. Péan, S. Berger, N. Caud, Y. Chen, L.Goldfarb, M. I. Gomis, M. Huang, K. Leitzell, E. Lonnoy, J.B.R. Matthews, T. K. Maycock, T. Waterfield, O. Yelekçi, R. Yu and B. Zhou (eds.)]. Cambridge University Press. In Press.

Iturbide, M., Gutiérrez, J.M., Alves, L.M., Bedia, J., Cerezo-Mota, R., Cimadevilla, E., Cofiño, A.S., Di Luca, A., Faria, S.H., Gorodetskaya, I.V., Hauser, M., Herrera, S., Hennessy, K., Hewitt, H.T., Jones, R.G., Krakovska, S., Manzanas, R., Martínez-Castro, D., Narisma, G.T., Nurhati, I.S., Pinto, I., Seneviratne, S.I., van den Hurk, B. and Vera, C.S. (2020) An update of IPCC climate reference regions for subcontinental analysis of climate model data: definition and aggregated datasets. Earth System Science Data, 12(4), 2959–2970. https://doi.org/10.5194/essd-12-2959-2020

(12) The conclusion section is not well written. What new things the manuscript provide? It is recommended to summarize from two aspects: method and finding. How well does the new method/perspective works and what is the scientific value of the results in this study?

Response: Thank you for the suggestion. We have re-written the conclusion section from two aspects of methods and findings. The new methods, findings, and implications have been added. Please find the revised text below.

In this study, the historical TWSA series over global land during 1985-2014 was calculated from the ensemble-mean of nine model outputs including three each from GHMs (VIC, WGHM, PCR-GLOBWB), LSMs (Noah, CLSM, CPC), and GRACE reconstructions (CSR, JPL, GSFC). Future TWSA projections from 2070 to 2100 under SSP126, SSP245, and SSP585 scenarios were derived from the average of eight selected CMIP6 GCMs after bias-correction using GRACE observations. Subsequently, TWS-DSI was estimated to detect the long-term trends in dryness/wetness in the past and future periods. Further, the DDWW paradigm has been re-examined with a significance level of 0.05 from the perspective of terrestrial water storage change. The uncertainty sourced from different choices of models, methods, and confidence levels has been discussed systematically. The new findings were summarized as follows.

(1) During the historical period, 32.9% and 22.1% of land area present significant ($p<0.05$) drying and wetting trends, respectively. During the future period under climate change, the proportion of drying areas with a significant slope increases from SSP126 (23.6%) to SSP585 (30.1%) scenario. Similar change is detected in the percentage with significant wetting trends, which reaches 15.7%, 17.4%, and 23.4% under SSP126, SSP245, and SSP585 scenarios, respectively.

(2) A total of 28.1% of the global land area shows the DDWW paradigm valid, in which 16.7% and 11.4% of the area is drying and wetting, respectively during the period 1985-2014. 23.3% of the area, however, shows the opposite pattern like "dry gets wetter" (DW, 8.4%) or "wet gets drier" (WD, 14.9%), respectively. The proportion of areas supporting the DDWW paradigm is 18.2%, 17.4%, and 20.7% under SSP126, SSP245, and SSP585 scenarios, respectively. Alternatively, the

area opposing the DDWW paradigm achieves 17.9%, 22.4%, and 28.5%, respectively.

(3) The ensemble-mean of TWSA generally compares better with GRACE observations during 2002-2014 than the individual solution, especially for the eight bias-corrected CMIP6 GCMs. Independent experiments based on the individual TWSA dataset suggest that the divergent choices of data source might lead to reasonable overestimations (CSR mascon) and underestimations (WGHM and VIC) for both the DDWW-agreed and DDWW-opposed patterns. Moreover, the use of distinctive GCMs suggests slightly overrated (GFDL-ESM4) and underrated (CanESM5) percentages of DDWW-pro and DDWW-con area in the future under multiple emission scenarios.

(4) Sensitivity analysis on different choices of significance levels from 0.01 to 0.1 indicate similar patterns, in which 22.2% (17.1%) of the land area supports (opposes) the DDWW theory historically under the 0.01 level, and the DDWW-validated regions account for the 30.6% of total area with 25.4% of land agreeing with the opposite hypothesis under the 0.1 level. Such consistency is also evidenced from the projected TWS-DSI in the future under various scenarios.

New insights from the TWSA perspective highlight that the widely-used DDWW paradigm is still challenging in both historical and future periods under climate change. In addition, our developed ensemble-mean method can effectively and efficiently alleviate the uncertainty sourced from different data sources, implying an alternative way to assess the TWSA variations over major basins globally. The regional aggregation of our study based on IPCC SREX reference regions can provide important inferences for decision-makers and stakeholders for the sustainable management and utilization of water resources under global change.

(13) Figure 1: Which method was used to calculate the slopes in the left panel? Which method was used to analyse the significance of trends? Which level?

Response: We estimated the long-term trends in TWS-DSI during 1985-2014 using the linear regression method, and the significance of trend values is evaluated using the t-test at a 5% significance level (Greve et al., 2014). The area having a significant trend of increasing/decreasing TWS-DSI is considered undergoing wetting/drying, otherwise it is defined as an uncertain region. We have added the explanation in the method section of the revised manuscript as follows:

Long-term trends in TWS-DSI were estimated using the linear regression method and the significance of trend values are evaluated using the t-test at a 5% significance level (Greve et al., 2014). The area having a significant trend of increasing/decreasing TWS-DSI is considered undergoing wetting/drying, otherwise, it is defined as an uncertain region.

**Reference:**

Greve, P., Orlowsky, B., Mueller, B., Sheffield, J., Reichstein, M., Seneviratne, S.I., 2014. Global assessment of trends in wetting and drying over land. Nat. Geosci. 7, 716–721. https://doi.org/10.1038/NGEO2247

(14) Figure 2: What does the fan shape in the map means?

Response: Thank you for the comment. The fan shape represents the regional proportion of area with different trends. The figure caption has been revised as Figure R10 below.

[Figure]

Figure R10 Global distribution of the long-term trends in TWS-DSI in 43 selected IPCC SREX regions during the (a) historical (1985-2014) and future (2071-2100) period under (b) SSP126, (c) SSP245, and (d) SSP585 scenarios. Note: The pie chart represents the regional proportion of area with different trends. "D" and "W" indicate regions with drying and wetting trends, respectively. Please refer to Figure S1 (added below) for abbreviations of the IPCC SREX regions.

(15) Figure 4: I cannot figure the fan shapes and their meaning clearly.

Response: Thank you for your comment. The figure has been revised accordingly as Figure R11:

[Figure]

Figure R11 Global assessment of the DDWW paradigm in 43 selected IPCC SREX regions during the (a) historical (1985-2014) and future (2071-2100) period under (b) SSP126, (c) SSP245, and (d) SSP585 scenarios. Note: The light grey colour represents an insignificant pattern. The pie chart represents the regional proportion of area with different patterns to the total area with significant ($p<0.05$) patterns. "D" and "W" indicate regions with drying and wetting trends, respectively. DD indicates the dry gets drier; DW indicates the dry gets wetter; WW indicates the wet gets wetter; WD indicates the wet gets drier; TD indicates the transition gets drier; TW indicates the transition gets wetter. Please refer to Figure S1 (added below) for abbreviations of the IPCC SREX regions.

[Figure]

Figure S1. Location of the 43 selected Special Report on Managing the Risks of Extreme Events and Disasters to Advance Climate Adaptation (SREX) regions from the Intergovernmental Panel on Climate Change (IPCC) Sixth Assessment Report (AR6). The regional abbreviations are listed in Table S3.

---

## Author Comment (AC2)

We thank the reviewer for his/her time in reading our manuscript and detailed comments on our manuscript. Point-by-point replies to the comments or suggestions made can be found below.

**Reviewer #2:** The authors present a re-examination of the dry gets drier, and wet gets wetter paradigm over global land, based on terrestrial water storage estimates from different sources. They make use of GRACE reconstructions, global hydrological models, and land surface models, as well as CMIP6 models for the future perspective. They conclude that the DDWW paradigm is challenged both in the historical period but also in the future. I think that the topic is interesting and fits well the journal, and the use of the complete terrestrial water storage for the analysis of the DDWW paradigm adds another perspective compared to previous studies. However, the paper currently lacks some important information and has a substantial methodological issue which requires major revision.

Response: We thank the reviewer for recognizing the potential of the manuscript's new perspective and his/her detailed suggestions for improvement. All the concerns raised have been addressed in the revised manuscript. We hope the modified text along with the supplementary analyses and discussions will put forward the results in a much more robust way.

**Major comments:**

(1) The use of percentage of grid cells for the presentation of many of the results is not appropriate and hinders the proper interpretation. It's necessary to present the corresponding numbers as percentage per land area (i.e., by weighing the grid boxes according to their effective km$^2$ area) in order not to give excessive weight to higher latitudes. This will most likely have impacts on the overall conclusions of the paper.

Response: Thank you very much for pointing out this mistake. As suggested, we have performed the re-calculation based on the actual area instead of the number of grid cells to evaluate the DDWW paradigm over global land (see Figure R1). The main findings have been summarized in the conclusion section of the revised manuscript.

[Figure]

Figure R1. Spatial distribution of actual area of the 1° grid cells over global land.

The revised conclusions of the manuscript are as below (same as section 4 in the revised manuscript):

In this study, the historical TWSA series over global land during 1985-2014 was calculated from the ensemble-mean of nine model outputs including three each from GHMs (VIC, WGHM, PCR-GLOBWB), LSMs (Noah, CLSM, CPC), and GRACE reconstructions (CSR, JPL, GSFC). Future TWSA projections from 2070 to 2100 under SSP126, SSP245, and SSP585 scenarios were derived from the average of eight selected CMIP6 GCMs after bias-correction using GRACE observations. Subsequently, TWS-DSI was estimated to detect the long-term trends in dryness/wetness in the past and future periods. Further, the DDWW paradigm has been re-examined with a significance level of 0.05 from the perspective of terrestrial water storage change. The uncertainty sourced from different choices of models, methods, and confidence levels has been discussed systematically. The new findings were summarized as follows.

(1) During the historical period, 32.9% and 22.1% of land area present significant ($p<0.05$) drying and wetting trends, respectively. During the future period under climate change, the proportion of drying areas with a significant slope increases from SSP126 (23.6%) to SSP585 (30.1%) scenario. Similar change is detected in the percentage with significant wetting trends, which reaches 15.7%, 17.4%, and 23.4% under SSP126, SSP245, and SSP585 scenarios, respectively.

(2) A total of 28.1% of the global land area shows the DDWW paradigm valid, in which 16.7% and 11.4% of the area is drying and wetting, respectively during the period 1985-2014. 23.3% of the area, however, shows the opposite pattern like "dry gets wetter" (DW, 8.4%) or "wet gets drier" (WD, 14.9%), respectively. The proportion of areas supporting the DDWW paradigm is 18.2%, 17.4%, and 20.7% under SSP126, SSP245, and SSP585 scenarios, respectively. Alternatively, the area opposing the DDWW paradigm achieves 17.9%, 22.4%, and 28.5%, respectively.

(3) The ensemble-mean of TWSA generally compares better with GRACE observations during

2002-2014 than the individual solution, especially for the eight bias-corrected CMIP6 GCMs. Independent experiments based on the individual TWSA dataset suggest that the divergent choices of data source might lead to reasonable overestimations (CSR mascon) and underestimations (WGHM and VIC) for both the DDWW-agreed and DDWW-opposed patterns. Moreover, the use of distinctive GCMs suggests slightly overrated (GFDL-ESM4) and underrated (CanESM5) percentages of DDWW-pro and DDWW-con area in the future under multiple emission scenarios.

(4) Sensitivity analysis on different choices of significance levels from 0.01 to 0.1 indicate similar patterns, in which 22.2% (17.1%) of the land area supports (opposes) the DDWW theory historically under the 0.01 level, and the DDWW-validated regions account for the 30.6% of total area with 25.4% of land agreeing with the opposite hypothesis under the 0.1 level. Such consistency is also evidenced from the projected TWS-DSI in the future under various scenarios.

New insights from the TWSA perspective highlight that the widely-used DDWW paradigm is still challenging in both historical and future periods under climate change. In addition, our developed ensemble-mean method can effectively and efficiently alleviate the uncertainty sourced from different data sources, implying an alternative way to assess the TWSA variations over major basins globally. The regional aggregation of our study based on IPCC SREX reference regions can provide important inferences for decision-makers and stakeholders for the sustainable management and utilization of water resources under global change.

(2) The choice of the eight CMIP6 models is not transparent. Why didn't the authors choose a larger model ensemble based on the CMIP6 archive? Based on which criteria were these eight models selected?

Response: We thank you for the instructive comment. We have clarified the criteria to select these eight CMIP6 models in the Data section of the revised manuscript:

We chose these eight models out of the 34 CMIP6 models because, as we write, they are the only models for which TWSA results are available in both the historical and future period under multiple emission scenarios (see Table S1). The CMIP6 TWSA represents the sum of total soil moisture and snow equivalent water, which has been comprehensively validated, with embedded uncertainties, over global major river basins compared with the GRACE data (Freedman et al., 2014; Wu et al., 2021).

**Reference:**

Freedman, F.R., Pitts, K.L., Bridger, A.F.C., 2014. Evaluation of CMIP climate model hydrological output for the Mississippi River basin using GRACE satellite observations. J. Hydrol. 519, 3566–3577. https://doi.org/10.1016/j.jhydrol.2014.10.036.
Wu, R.-J., Lo, M.-H., Scanlon, B.R., 2021. The annual cycle of terrestrial water storage anomalies in CMIP6 models evaluated against GRACE data. J. Clim. 34, 8205–8217. https://doi.org/10.1175/JCLI-D-21-0021.1

(3) Also, given the large uncertainties between the CMIP6 models on the one hand, but also within

the DATASET ensemble (cf. Fig. S8), the impact of the applied ensemble mean approach on the results should be discussed in more detail.

Response: Thank you for the comment. We have discussed the results utilizing the individual model outputs and clarified the rationale for employing the ensemble mean approach. Moreover, we have added discussions on the uncertainty and implications of our study in the updated manuscript, as below (same as newly added section 3.3 of the revised manuscript).

Each ensemble member of the DATASET has embedded uncertainties inherently originating from one or more of forcing variables, simplified assumptions of complex processes in the models and their physical structure, retrieval algorithms, and systematic biases, which inevitably have propagated to the results presented herein. For example, the original GRACE mascon observations contain the measurement error and signal leakage at gridded scale, and they further spread into the reconstruction of TWSA when training via the statistical methods (Humphrey and Gudmundsson, 2019; Li et al., 2021a). Unlike observed GRACE and reconstructed GRACE-like data, the TWSA simulations from the GHMs, LSMs, and GCMs are featured by incomplete representation (Table S2). They are generally based on the simplified hydrological processes, resulting in the missing some of the TWSA components. For example, the widely used Noah model lacks the surface water and groundwater storage in TWSA, and all the GCMs can only simulate the snow water and soil moisture within a limited depth from 2 to 10 m below land surface (Xiong et al., 2021b; Wu et al., 2021). Moreover, the eight CMIP6 GCMs are forced with the future projections of many meteorological variables such as precipitation and air temperature, which have been reported showing underestimation or overestimation nearly over the global land (Eyring et al., 2016; Kim et al., 2020). Despite employing bias correction with GRACE data, propagated uncertainty from the forcing and models can influence the accuracy of TWSA simulations (Xiong et al., 2022). Although it is challenging to explicitly attribute and quantify these uncertainties in absence of a true reference observation dataset, the ensemble averaging method has been used to integrate the multi-source TWSA data. The global distributions of NRMSE between GRACE observations and each ensemble member and their mean during April 2002-December 2014 show improved performance of the latter (Figure S7). Three GRACE reconstructions present relatively lower error than the GHMs and LSMs, especially in the high-latitude northern hemisphere where snow, ice, and glaciers contribute more to TWS than other regions, which is not considered in most of the global models. The ensemble-mean solution illustrates the reasonably good accuracy with the NRMSE generally below 0.2, highlighting the reduced uncertainty compared with the individual solution. It is not surprising that the GRACE reconstructions compare better than other data because they are directly calibrated with the GRACE measurements during 2002-2017. While their performances need more validation beyond the GRACE era (i.e., prior to April 2002 and during July 2017-June 2018). Similar patterns are also evident from the probability density functions of NRMSE, of which there is an overall negative deviation in the ensemble-mean relative to other solutions except for the CSR

reconstruction (see Figure S8). This outperformance of the ensemble dataset is ascertained by the increased correlation and decreased standard deviation as shown by the Taylor diagram (Figure S8). In addition, the comparison between GCM-modelled and GRACE TWSA in the past (April 2002-December 2014) is conducted (see Figure S9). The spatial distributions clearly show that the ensemble-mean of eight GCMs outperforms each member globally, particularly in Australia, southern Africa, and North America. An overall decrease in NRMSE is observed according to the probability density functions, which is also detected from the Taylor diagram results (see Figure S10).

Further, we carried out an independent analysis at the individual member level (see Figure S11). For the historical period, a clear overestimation of the CSR reconstructions is detected with 42.4% of the area agreeing with the DDWW pattern, and 36.6% showing the opposite situation. Moreover, the modelled results from VIC and WGHM illustrate the underestimation of the area validating the DDWW paradigm, reaching 15.6% (WGHM) and 12.2% (VIC), respectively. Their proportion with the opposite DDWW paradigm is 10.2% (WGHM) and 17.8% (VIC), respectively. Therefore, it can be concluded that the differences among different members of DATASET limitedly affect the evaluation of the DDWW during the historical period. In the future, the GFDL-ESM4 model presents overestimation but the IPSL-CM6A and CanESM5 models have underestimation for different percentages compared with the ensemble-mean. Specifically, the area dominated by the DDWW paradigm changes from 8.9% (CanESM5) to 21.9% (GFDL-ESM4), while that showing the opposite pattern ranges from 7.8% (CanESM5) to 14.8% (GFDL-ESM4) under the SSP126 scenario. For the SSP245 scenario, the DDWW-validated regions account from 7.4% (CanESM5) to 21.5% (GFDL-ESM4), the opposite pattern occurs over a range from 9.7% (CanESM5) to 16.0% (GFDL-ESM4) of land. The proportion supporting the DDWW paradigm varies from 10.4% (CanESM5) to 24.0% (GFDL-ESM4), while that presenting the opposite pattern ranges from 8.4% (CanESM5) to 22.3% (GFDL-ESM4) under the SSP585 scenario. Overall, the comparatively large difference among various models might source from unforced internal climate variability of distinctive CMIP6 members and different emission scenarios (Kumar et al., 2015).

Our choice of the significance level (i.e., 0.05) may also affect the rationale of the DDWW examination results, thus different significance levels are alternatively tested (see Figure S11). At a significance level of 0.01, 22.2% of land area agrees well with the DDWW theory, while the 17.1% of area illustrates the opposite pattern during the period 1985-2014. As for the 0.1 significance level, the DDWW-validated regions account for 30.6% of the total area, with 25.4% of land agreeing with the opposite hypothesis. In the future period, a similar pattern is discovered that both DDWW-confirmed and DDWW-opposed regions are increasing on account of the enhancement of projected strength of radiative forcing, with the reduction of the area showing insignificant trends in wetting and drying. However, the magnitudes of results at the 0.01 significance level are generally lower than that at the 0.1 significance level due to the different thresholds of detected trends in drying and wetting.

Despite the multisource uncertainties, our study can provide important implications for the long-term trends in dryness/wetness over global land in the past and future from the perspective of TWSA. Compared with other widely used indexes that are purely derived from the hydrometeorological variables (e.g., SPI, SPEI, and PDSI) or incorporate a single component of the TWSA (e.g., SSI, SGI, and SRI), our developed TWS-DSI describes the overall status of the land system, which is jointly influenced by different components including soil moisture, river runoff, and groundwater that play different roles in the hydrological cycle (Tapley et al., 2019). The new insights benefit the comprehensive evaluation of terrestrial conditions over regions where some parts of TWSA (e.g., groundwater storage and snow water) have been rapidly depleting due to intensive human activities and warming climate worldwide, including the Qinghai-Tibet Plateau and northwest India (Rodell et al., 2009; Xing et al., 2021). Furthermore, the projected changes in global TWSA and associated TWS-DSI improve our understanding of the large-scale hydrological response under climate change, particularly in regions with strong human interventions such as the south and east of Asia. Despite the magnitude bias from satellite products, simulations of LSMs and GHMs, and GCMs projections, the ensemble averaging method has presented an effective and efficient ability to alleviate the multi-source uncertainty, which can be further applied over data-sparse areas globally with limited in-situ observations like Africa and central Asia. In addition, the regional aggregation of the analysis based on the IPCC AR6 SREX references regions can supply valuable inferences for policymakers and stakeholders for better water resources management in a changing environment (Iturbide et al., 2020).

**Specific comments:**
(1) Line 20: "and freshwater availability" instead of "and fresh availability"

Response: We regret the error. We have revised the sentence as follows:

The hydrological conditions of the land surface have experienced considerable changes due to climate change and anthropogenic interventions, exerting a tremendous impact on regional agriculture, ecological environment, and freshwater availability (Shugar et al., 2020; Gampe et al., 2021).

(2) Line 83/84: Which of the available meteorological forcings for these reconstructions did you consider (MSWEP, GSWP3 or ERA5)? Did you take the mean over the three forcings for each of the two calibrations?

Response: Yes, we have opted for the ensemble mean of these reconstructions to avoid implicit biases to a single meteorological forcing an explanation of the GRACE reconstructions used in our study has been added as follows:

We note that the ensemble-mean of the NASA JPL and GSFC reconstructions forced with the multisource weighted-ensemble precipitation (MSWEP), the Global Soil Wetness Project Phase 3 (GSWP3), and the European Centre for Medium-Range Weather Forecasts reanalysis (ERA5) datasets have been taken, respectively. The CSR reconstruction is derived from four kinds of meteorological variables (e.g., precipitation and 2 m temperature) and three kinds of hydrological

variables (e.g., soil moisture and runoff) (Li et al., 2021).

(3) Line 125: Based on which criteria did you select these 8 models?

Response: Thank you for the suggestion. The selection criterion is constrained by the availability of the data. Please also see our response to Major comment 2 above for details.

(4) Line 136: "to match the observed data" instead of "to match the observed results"

Response: As suggested, we have revised the sentence.

(5) Line 144 (Equation 1): The dash in TWS – DSI could be confused with a "minus". I suggest changing it to an "_" or at least a short dash.

Response: Thank you for the suggestion. We have revised the representation of TWS-DSI in Equation 1 of the manuscript using a short dash.

(6) Line 152: "all the land area except for the Greenland and Antarctica"

Response: Following your constructive comment, we have revised this sentence as follows:
A total of 43 regions are selected based on the Special Report on Managing the Risks of Extreme Events and Disasters to Advance Climate Adaptation (SREX) from Intergovernmental Panel on Climate Change (IPCC) Sixth Assessment Report (AR6), which covers all the land area except for the Greenland and Antarctica (see Figure S1 below).

[Figure]

Figure S1. Location of the 43 selected Special Report on Managing the Risks of Extreme Events and Disasters to Advance Climate Adaptation (SREX) regions from the Intergovernmental

(7) Line 157: "CMIP6 archive" instead of "CMIP6 achieve"

Response: We have rectified this typo.

(8) Line 168-170: Why poorer performance when NRMSE is lower?

Response: Thank you so much for the comment. It is an inadvertent error. We have revised the sentence as follows:

Most mid-latitude regions like the WCE, EEU, WSB, ESB, and RFE present relatively lower NRMSE (0-0.1) between GRACE and DATASET, suggesting better performance than that in the NZ, ECA, NEU, and NEN.

(9) Line 180: "greater" instead of "slighter"? The fluctuations of CMIP6 are larger than the ones of DATASET (Fig. S4).

Response: Considering the fluctuations of CMIP6 are larger than the ones of DATASET, we have revised this statement according to your suggestion as follows:

Moreover, the fluctuation range of CMIP6 data is generally greater than the DATASET, highlighting the considerable uncertainty sourced from different forcing variables and model parameterizations.

(10) Line 180: "the effective bias correction performance" Why effective? CMIP6 deviates more from GRACE than DATASET.

Response: We thank you for the comment. This sentence has been modified in the updated version of the manuscript.

(11) Line 212: Here and throughout the results the use of percentage of grid cells is not appropriate and needs to be changed to percentage of land area for proper interpretation.

Response: As suggested, we have re-calculated all the results based on the actual area instead of the number of grid cells and updated the results through the manuscript. Subsequently, the relevant text has been modified throughout the manuscript to reflect the changes.

(12) Figure 2: Nice plot, however quite crowded. The region names are often barely readable. You could just refer to the Supplementary Figure 1 for the definition and naming of the regions. The same applies to Figure 4.

Response: Following your constructive suggestion, we have removed the region names in Figure 2 and Figure 4 of the original manuscript, as shown in Figure R2 and R3 below.

[Figure]

Figure R2 Global distribution of the long-term trends in TWS-DSI in 43 selected IPCC SREX regions during the (a) historical (1985-2014) and future (2071-2100) period under (b) SSP126, (c) SSP245, and (d) SSP585 scenarios. Note: The pie chart represents the regional proportion of area with different trends. "D" and "W" indicate regions with drying and wetting trends, respectively. Please refer to Figure S1 (added below) for abbreviations of the IPCC SREX regions.

[Figure]

Figure R3 Global assessment of the DDWW paradigm in 43 selected IPCC SREX regions during the (a) historical (1985-2014) and future (2071-2100) period under (b) SSP126, (c) SSP245, and (d) SSP585 scenarios. Note: The light grey colour represents an insignificant pattern. The pie chart represents the regional proportion of area with different patterns to the total area with significant (p<0.05) patterns. "D" and "W" indicate regions with drying and wetting trends, respectively. DD indicates the dry gets drier; DW indicates the dry gets wetter; WW indicates the wet gets wetter; WD indicates the wet gets drier; TD indicates the transition gets drier; TW indicates the transition gets wetter. Please refer to Figure S1 (added below) for abbreviations of the IPCC SREX regions.

(13) Line 234: There's no stippling on these figures? Please revise the caption and also explain the meaning of the pie charts.

Response: Thank you for this valuable comment. We have revised the caption and added an explanation of these figures as Figure R2 above.

(14) Line 235: Same title as for Section 3.1. I guess this is an oversight.

Response: We apologize for this oversight. We have replaced the title for section 3.2 with "Assessment of the DDWW Paradigm".

(15) Line 257 and following: I assume these percentages are again based on the grid cells only, not based on the actual area?

Response: Thank you again for the kind reminder. We have updated the manuscript using the newly calculated results based on the actual area instead of the number of grid cells.

(16) Conclusion: The conclusions need to be extended. What's new compared to previous studies? What are the implications?

Response: We thank you for the enlightening suggestion. The conclusions have been systematically extended and itemized for better comprehension in the new version of the manuscript. Please find the modified conclusions in Major comment 1.